# Vilya, a component of the recombination nodule, is required for meiotic double-strand break formation in *Drosophila*

Cathleen M Lake[1], Rachel J Nielsen[1], Fengli Guo[1], Jay R Unruh[1], Brian D Slaughter[1], R Scott Hawley[1,2]*

[1]Stowers Institute for Medical Research, Kansas City, United States; [2]Department of Molecular and Integrative Physiology, Kansas University Medical Center, Kansas City, United States

**Abstract** Meiotic recombination begins with the induction of programmed double-strand breaks (DSBs). In most organisms only a fraction of DSBs become crossovers. Here we report a novel meiotic gene, *vilya*, which encodes a protein with homology to Zip3-like proteins shown to determine DSB fate in other organisms. Vilya is required for meiotic DSB formation, perhaps as a consequence of its interaction with the DSB accessory protein Mei-P22, and localizes to those DSB sites that will mature into crossovers. In early pachytene Vilya localizes along the central region of the synaptonemal complex and to discrete foci. The accumulation of Vilya at foci is dependent on DSB formation. Immuno-electron microscopy demonstrates that Vilya is a component of recombination nodules, which mark the sites of crossover formation. Thus Vilya links the mechanism of DSB formation to either the selection of those DSBs that will become crossovers or to the actual process of crossing over.

*For correspondence: rsh@stowers.org

**Competing interests:** The authors declare that no competing interests exist.

## Introduction

Meiosis is a specialized form of cell division that reduces the number of chromosomes in germ cells by half. This is achieved by coupling one round of DNA replication with two rounds of chromosome segregation. During the first meiotic division, homologous chromosomes segregate away from each other. At the second (mitosis-like) meiotic division sister chromatids segregate from each other, producing four meiotic products. Successful completion of the first meiotic division requires the proper completion of several key events, each of which must occur at a specific time and place during prophase. For instance, programmed double-strand breaks (DSBs), required for the initiation of meiotic recombination, are spatially and temporally controlled. Failure to initiate recombination, to create the correct number of DSBs, or to position the DSBs properly can lead to aneuploidy (*Murakami and Keeney, 2008*), which in humans can result in disorders of chromosome number such as Down, Klinefelter, or Turner syndrome.

The reason a failure in initiating recombination induces chromosome missegregation is because a subset of DSBs are repaired into crossovers, and, in most cases, it is the physical linkage (chiasmata) of the homologs by crossovers that ensures chromosomes segregate properly at the first meiotic division (*Page and Hawley, 2003*). Crossovers are formed within the context of the synaptonemal complex (SC), a highly conserved proteinaceous structure formed between homologs during early meiotic prophase (*Zickler and Kleckner, 1999*; *Page and Hawley, 2004*). The SC consists of two lateral elements (LEs) and a central region that contains both the central element (CE) and transverse filament (TF) proteins. Although crossover formation almost universally requires the presence of SC, the degree to which DSB formation depends on SC formation, or vice versa, differs between

**eLife digest** DNA in animal cells is arranged into structures called chromosomes. Usually, a cell divides in such a way that both daughter cells inherit a complete set of chromosomes. However, the sex cells (sperm and egg cells) are formed in a different process – called meiosis – that results in these cells having only half the number of chromosomes that the parent cell had. This ensures that when animals reproduce, an egg cell and a sperm fuse together to make a new cell that contains a full set of chromosomes.

During meiosis, sections of chromosomes are rearranged so that the sperm and egg cells will end up with different combinations of the DNA inherited from the animal's mother and father. Matching chromosomes from the mother and father pair up with each other and the DNA - which is made of two strands - breaks at precise locations throughout the chromosomes. Then, sections of DNA around the double-strand breaks are exchanged between the matching chromosomes by a process known as crossover. An incorrect number of double-strand breaks, or a failure to position them properly, can lead to genetic abnormalities like Down's syndrome, in which cells contain the wrong number of chromosomes.

Cells tightly regulate the formation of double-strand breaks, but in most organisms the number of breaks formed exceeds the number of crossovers. This implies that there must be a process that selects certain double-strand breaks to form crossovers. Although fruit flies are often used as a model to study animal cells, we do not know how they select which double-strand breaks will form crossovers.

Lake et al. studied meiosis in fruit flies and identified a new protein called Vilya that is required for double-strand breaks to form. This protein is similar to a group of "Zip3-like" proteins that act on double-strand breaks in other animals. Vilya is found at the double-strand break sites that go on to form crossovers and interacts with a small protein called Mei-P22, which is known to be involved in the formation of double-strand breaks.

Lake et al.apos;s findings show that Vilya links the process of forming double-strand breaks to the selection of the breaks that will undergo crossover. Future studies will focus on understanding the molecular details of how Vilya works.

organisms. In yeast (*Roeder, 1997*), SC formation is dependent on DSBs, as DSB sites appear to be the location for the initiation of SC synthesis (*Chua and Roeder, 1998*); and in mammals, SC formation between homologs is dependent on DSB formation (*Baudat et al., 2000 Romanienko and Camerini-Otero, 2000*). However, in flies (*McKim et al., 1998*; *Jang et al., 2003*) and worms (*Dernburg et al., 1998*), SC formation is not dependent on DSB formation, and in fact, DSBs are formed after full-length SC is constructed. Moreover, in the absence of SC formation in flies, DSB formation in the oocyte is significantly reduced (*Mehrotra and McKim, 2006*; *Collins et al., 2014*).

Although DSBs are induced by the evolutionarily conserved topoisomerase-like protein Spo11 (*Keeney et al., 1997*), many poorly conserved accessory proteins have been identified that are required either to facilitate the formation of the DSBs themselves or position the DSBs within the euchromatin (*de Massy, 2013*). Indeed, the process of DSB formation is tightly controlled, both in terms of DSB number and position. Recently, a feedback mechanism has been proposed that links the process of DSB repair to the DSB formation process in both yeast and worms (*Rosu et al., 2013*; *Thacker et al., 2014*). In addition, the position of DSBs within the genome is nonrandom, and in many organisms is often controlled by specific sequence motifs that create recombinational hotspots (*de Massy, 2013*).

In most organisms the number of DSBs far exceeds the number of crossover events (*de Massy, 2013*). For example, the ratio of DSBs to crossovers is 10 to one in mice (*Moens et al., 2002*), while in flies there are at least three times more DSBs than there are crossovers (*Lindsley et al., 1977*; *Mehrotra and McKim, 2006*). Therefore, there must be a selection process that differentiates those DSBs that become crossovers from those that will be repaired by processes that create noncrossover gene conversions. Recent studies have identified components of the multistep process that selects those DSBs that will become crossover-competent DSBs. These steps appear to be controlled by the ever-growing Zip3 family of proteins and their regulators.

Zip3 was first identified in yeast (*Ouspenski et al., 1999*; *Agarwal and Roeder, 2000*), and homologs have now been identified in many other model organisms. Recently it has been suggested that there are two subgroups within the Zip3 family: the Zip3/RNF212 group and the Hei10 group (*Chelysheva et al., 2012*; *De Muyt et al., 2014*). All the members within both subgroups are required for the formation of crossovers and are similar in terms of protein structure; they contain a RING-type zinc finger domain, an internal coiled-coil domain, and a C-terminal domain that tends to be serine rich (*Reynolds et al., 2013*). However, not all organisms possess members of both subgroups. Both budding yeast (*Agarwal and Roeder, 2000*) and worms (*Jantsch et al., 2004*; *Bhalla et al., 2008*) are predicted to carry only a single member of the Zip3/RNF212 group, whereas the Arabidopsis (*Chelysheva et al., 2012*), rice (*Wang et al., 2012*) and Sordaria (*De Muyt et al., 2014*) genomes are thought to encode only a member of the Hei10 group. The genomes of mammals, like humans (*Toby et al., 2003*; *Kong et al., 2008*) and mice (*Strong and Schimenti, 2010*; *Reynolds et al., 2013*; *Qiao et al., 2014*), appear to encode members from each subgroup.

The two subgroups display key differences in their overall enzymatic activity. Zip3/RNF212 group members appear to act solely as SUMO E3 ligases, whereas some members of the Hei10 group appear to possess both ubiquitin E3 ligase and SUMO E3 ligase activity. Yeast Zip3, which is required to regulate SUMO modification along meiotic chromosomes, has SUMO E3 ligase activity in vitro (*Cheng et al., 2006*), and genetic studies have implicated Zhp-3, the *C. elegans* Zip3 homolog, in the SUMO pathway as well (*Bhalla et al., 2008*). Conversely, human Hei10 has been shown biochemically to have ubiquitin E3 ligase activity in vitro (*Toby et al., 2003*). However, recent studies suggest that mouse Hei10 may also function as a SUMO E3 ligase (*Strong and Schimenti, 2010*). These observations suggest that the relationship between SUMOylation and ubiquitination of proteins in the vicinity of the DSB determines which DSBs become competent to crossover (*Qiao et al., 2014*).

Very little is known about how DSBs become crossover-competent DSBs in Drosophila. Prior to this study, homologs for most of the proteins required for this process in other organisms (Msh4/Msh5 (*Yokoo et al., 2012*), RNF212, Hei10 (*Strong and Schimenti, 2010*), Mlh1/Mlh3) had not been identified in Drosophila. In this manuscript we describe a new meiosis-specific gene that we have named *vilya*. Vilya is a Zip3-like RING-containing protein that is required for programmed DSB formation. Vilya interacts with another DSB accessory protein, Mei-P22, and these proteins localize to sites of DSBs as identified by the chromatin modification γH2AV (*Mehrotra and McKim, 2006*). When an epitope-tagged version of Vilya is expressed in the female germline, it shows a dynamic localization pattern that is dependent on DSB formation. In early pachytene, Vilya localizes both to the central region of the SC and to discrete foci. As the oocyte matures into early/mid-pachytene, Vilya is primarily found at discrete foci. The number and distribution of these foci along the euchromatic SC of each chromosome arm parallels the number and position of crossover events. Indeed, we show that Vilya is a component of recombination nodules (RNs) by immuno-electron microscopy (immuno-EM), making it the first RN protein component identified in Drosophila. We speculate that Vilya has functions that have recently been described for several members of the Zip3 group, such as DSB fate determination and crossover formation.

## Results

### Meiosis in Drosophila

Drosophila females provide an excellent system to analyze the progression of very early events of the first meiotic division because egg chambers within each ovariole of the ovary are arranged according to developmental age (*King et al., 1956*). *Figure 1A* shows a schematic of the Drosophila germarium, which is the structure at the very tip of the ovariole and is where meiosis begins. In region 1, the germline stem cell (GSC) divides to produce a cystoblast, which undergoes four rounds of incomplete cell division to produce a 16-cell interconnected cyst. These early divisions are known as the premeiotic divisions. Known components of the Drosophila SC, which are thought to be exclusively on meiotic chromosomes, associate with centromeres in the early premeiotic divisions and are required for the pairing and clustering of centromeres that begins at the eight-cell cyst (*Takeo et al., 2011*; *Tanneti et al., 2011*; *Christophorou et al., 2013*).

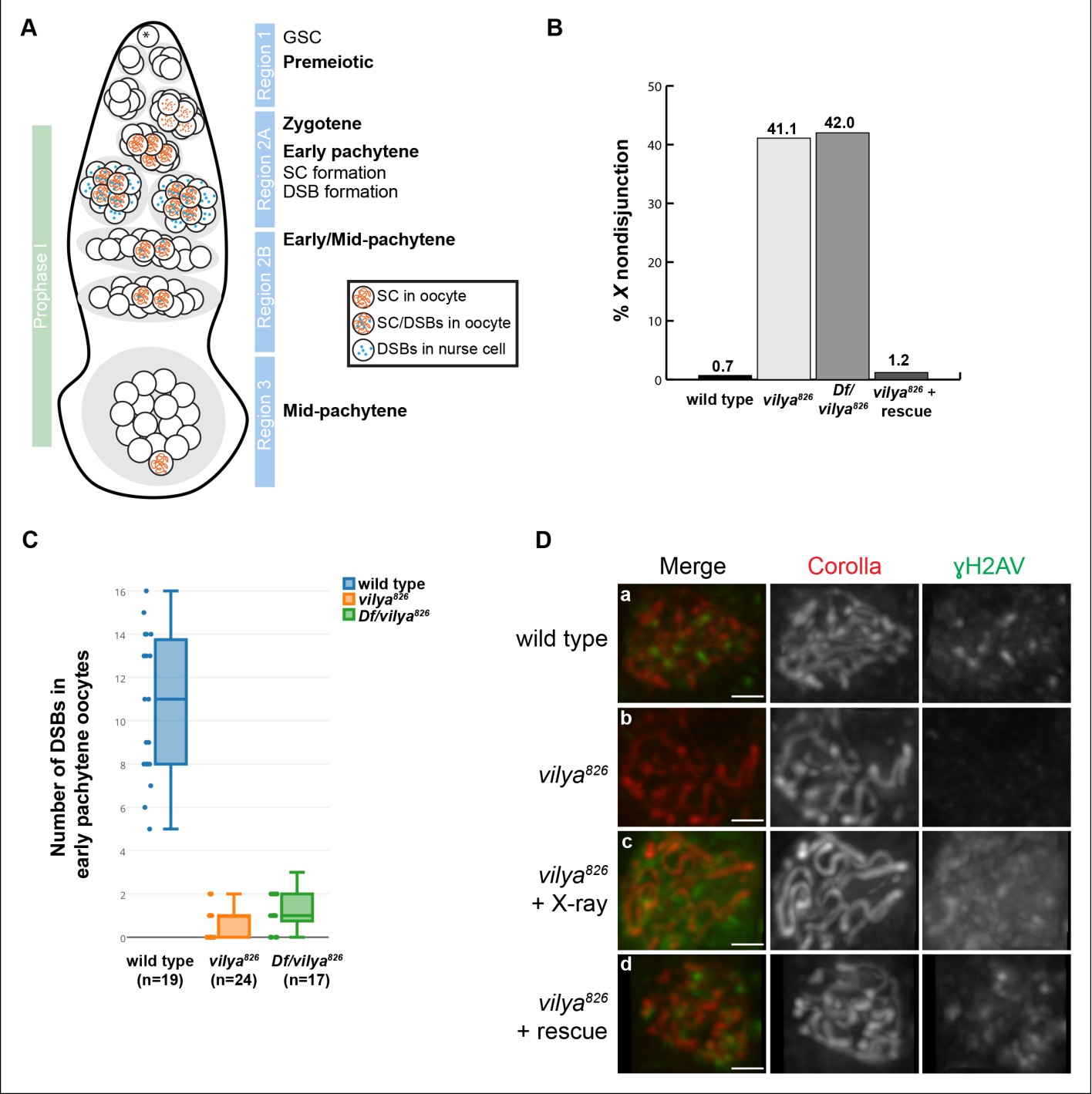

**Figure 1.** *vilya* encodes a RING domain-containing protein required for DSB formation. (**A**) Schematic diagram of a germarium showing the timing of SC and DSB formation. (**B**) *vilya*[826] homozygotes and Df/*vilya*[826] transheterozygotes cause high levels of *X* chromosome nondisjunction. The high level of *X* nondisjunction in *vilya*[826] is almost completely rescued by expressing *vilya*[3XHA] in the female germline. The deficiency that uncovers *vilya* used in the analysis was Df (1)ED6630. *vilya*[826] + rescue refers to the genotype *y w vilya*[826] *nos-Gal4/vilya*[826]; *PUASp-vilya*[3XHA]/+. Wild type and *vilya*[826] nondisjunction rates are from (***Collins et al., 2012***). (**C**) *vilya*[826] and Df/*vilya*[826] are defective in DSB formation in early pachytene oocytes as identified by an antibody against γH2AV and compared to wild type. DSBs in region 2A nurse cells are also significantly reduced in *vilya* mutants (see **Figure 1— figure supplement 4**). (**D**) Region 2A oocyte nuclei stained with Corolla (red) and γH2AV (green) in wild type, *vilya*[826], *vilya*[826] exposed to X-ray and *vilya*[826] + *vilya*[3XHA] germline rescue construct. Images are maximum intensity projections of deconvolved z-series through the selected nuclei. Scale bar, 1 μm.

*Figure 1. continued on next page*

*Figure 1. Continued*

The following figure supplements are available for Figure 1:

**Figure supplement 1.** *vilya, CG2709*, encodes a RING domain-containing protein.
**Figure supplement 2.** Centromere clustering and homolog pairing is not affected in *vilya*[826].
**Figure supplement 3.** C(3)G and Orb staining appears normal in *vilya*[826].
**Figure supplement 4.** Vilya plays a direct role in DSB formation in early pachytene.

Zygotene of prophase I begins in the 16-cell cyst, in region 2A, which is best defined by the presence of additional punctate SC staining throughout the euchromatin in up to four nuclei. As the cyst progresses in region 2A, it enters into pachytene where full-length SC is formed. In Drosophila, meiotic DSBs are formed by the Spo11 homolog, Mei-W68, after the SC is fully formed (***McKim and Hayashi-Hagihara, 1998***; ***Mehrotra and McKim, 2006***). DSBs can be visualized in Drosophila by the rapid phosphorylation of the histone 2A variant (γH2AV) at DSB sites that occur in all 16 nuclei within the cyst (in both the pro-oocytes and surrounding nurse cells) in region 2A (***Mehrotra and McKim, 2006***). As the cyst progresses into region 2B (early/mid-pachytene), only two nuclei have complete SC, and DSB numbers are reduced from those found in early pachytene. By region 3, or mid-pachytene, the oocyte nucleus has been selected and most of the γH2AV staining at DSB sites is removed, indicating that repair is either in progress or complete.

## Identification of the *vilya* mutant

A germline clone screen for EMS-induced meiotic mutations on the *X* chromosome produced a novel meiotic mutation, known initially as *mei-826,* that caused high levels of nondisjunction at the first meiotic division (***Collins et al., 2012***). This fully recessive mutation resulted in a C–T transition within a previously uncharacterized gene known as *CG2709* (***Figure 1—figure supplement 1A***) and is predicted to truncate the protein 24 amino acids from the end (R213STOP) (Materials and methods) (***Figure 1—figure supplement 1B***). We have named this gene *vilya* and have therefore subsequently renamed the mutant, *vilya*[826]. A transgene construct expressing a tagged version of the wild-type *vilya* gene (denoted *vilya*[3XHA]) in the germline fully rescued the chromosome segregation defect seen *in vilya*[826] homozygotes (***Figure 1B***). In addition, the meiotic nondisjunction phenotype of *vilya*[826] homozygotes was very similar to *Df(vilya)/vilya*[826] transheterozygotes, suggesting that *vilya*[826] is a null allele (***Figure 1B***).

## Vilya is a RING domain-containing protein

*vilya* is predicted to encode a protein with several identifiable domains. In the N-terminal region there is a $Cys_3HisCys_4$ Really Interesting New Gene (RING) domain (***Figure 1—figure supplement 1B,C***). RING domains are structural domains that bind two zinc cations and are typically found in E3 ligases (***Metzger et al., 2014***). In the middle of the protein there is a predicted coiled-coil domain (***Figure 1—figure supplement 1D***) (***Lupas et al., 1991***). Coiled-coil domains are often involved in protein–protein interactions and are commonly found in proteins that localize to the SC (***Sym et al., 1993***; ***Page and Hawley, 2004***; ***Smolikov et al., 2009***; ***Collins et al., 2014***). Additionally, the C-terminal region of Vilya is serine rich, with the last quarter of the protein being approximately 25% serines (***Figure 1—figure supplement 1E***). These characteristics are typical of members of the Zip3 protein family (***Reynolds et al., 2013***) (see Discussion).

## *vilya* is required for programmed DSB formation in early pachytene

Since *vilya*[826] causes very high levels of chromosome missegregation and encodes a protein with a potential coiled-coil domain, we asked whether this mutant was disrupting the early events in meiotic prophase. Specifically, we wondered if it affected SC formation or two processes that depend on the SC: the pairing and clustering of centromeres and the pairing of homologous chromosomes. We first assayed the processes of centromere clustering (***Figure 1—figure supplement 2A***) and homolog pairing (***Figure 1—figure supplement 2B***) in early pachytene nuclei. Unlike mutants that

fail to pair and/or cluster their centromeres properly and thus display greater than three centromere foci (*Takeo et al., 2011*), oocytes homozygous for *vilya*[826] showed no defects in centromere pairing/clustering when compared to wild type. Similarly, *vilya*[826] was not defective in euchromatic homolog pairing as assayed for the *X* chromosome by fluorescence in situ hybridization (FISH). Moreover, immunofluorescence analysis of early pachytene nuclei did not reveal defects in the ability of the SC protein Corolla to localize properly in *vilya*[826] germaria (*Figure 1D–a,b*). As well, we did not detect defects in timing or localization of the TF SC protein, C(3)G, or Orb, a cytoplasmic marker for oocyte determination (*Figure 1—figure supplement 3*). Taken together, we were unable to detect significant defects in any of the processes that occur prior to the initiation of DSBs.

However, the formation of DSBs, as assayed with an antibody recognizing γH2AV, was greatly reduced in *vilya*[826] and *Df/vilya*[826] oocytes. Specifically, we assayed the initiation of DSBs that occur in all nuclei in region 2A cysts within the germarium (see *Figure 1A*) by comparing the timing and presence of γH2AV foci of *vilya*[826] homozygotes and *Df/vilya*[826] transheterozygotes to that observed in wild type, in an SC mutant (*c(3)G*) that initiates DSBs albeit at reduced levels in the oocyte (*Mehrotra and McKim, 2006*), and in two DSB-defective mutants (*mei-W68* and *mei-P22*) (*Figure 1—figure supplement 4*). We found that unlike wild-type and *c(3)G* females, where γH2AV foci are readily observed in region 2A cysts, *vilya*[826] and *Df/vilya*[826] females show an almost complete absence of γH2AV staining, similar to *mei-W68* and *mei-P22*. These observations strongly suggest that *vilya*[826] and *Df/vilya*[826] oocytes are defective in DSB formation.

Immunofluorescence analysis of *vilya*[826] also reveals a severe failure to initiate programmed DSBs in oocytes in early pachytene compared to wild type (*Figure 1C* and *Figure 1D–a,b*). This defect is not caused by a delay in DSB formation, as no γH2AV foci were detected in later stages of pachytene in the germarium (*Figure 1—figure supplement 4*). The near complete absence of the γH2AV foci in *vilya*[826] oocytes was also not due to an inability of *vilya*[826] to modify the histone at DSB sites, as γH2AV was detected in *vilya*[826] oocytes when DSBs were artificially induced by X-ray treatment (*Figure 1D,C*). Finally, the failure to induce DSBs in *vilya* mutant females is solely due to the lack of functional Vilya because germline expression of *vilya*[3XHA] is able to rescue DSB formation (*Figure 1D–d*).

To rule out the possibility that the observed reduction of DSBs by this assay was not due to an increased rate of DSB repair in *vilya*[826] oocytes, we analyzed the ability of *vilya*[826] to rescue the defects associated with the DSB repair-deficient mutant, *okra* (homolog of yeast Rad54). In *okra* mutant-bearing oocytes, DSBs are left unrepaired, leading to the activation of a DNA damage checkpoint (*Ghabrial et al., 1998*). Activation of this checkpoint induces several observable phenotypes, which are bypassed by mutants that fail to form DSBs (*Ghabrial and Schupbach, 1999*; *Liu et al., 2002*; *Lake et al., 2011*). We examined the effect of the *vilya* mutant on two of these phenotypes. First, in *okra* mutant-bearing oocytes, the presence of unrepaired breaks leads to sterility (*Figure 2A*) (*Ghabrial et al., 1998*). In *vilya*[826]/+ oocytes, which carry one wild-type copy of *vilya*, homozygosity for *okra* causes near complete sterility, producing only 0.2 progeny per female on average. However, in the *vilya*[826] *okra* double mutant, the fertility was similar to *vilya*[826] alone, averaging 14.4 and 16.3 progeny per female, respectively, and the rate of *X* and *4th* chromosome nondisjunction in the double mutant was similar to the *vilya*[826] single mutant (*Figure 2A*). Therefore, the reduction of DSBs due to the *vilya* mutation resulted in the rescue of fertility (about 30% as fertile as wild type) caused by the *okra* mutation. Second, in *okra* mutant oocytes the presence of unrepaired DSBs results in an inability to form the spherical meiotic chromosome mass, known as the karyosome, in late pachytene (*Ghabrial et al., 1998*). In the presence of DSBs, but in the absence of repair, the karyosome structure is fragmented (*Figure 2—figure supplement 1*). *vilya*[826] was also able to rescue the karyosome defect seen in *okra* mutants (*Figure 2B* and *Figure 2—figure supplement 1*). Although we cannot rule out the possibility that DSBs are formed and repaired in a *rad54/okra*-independent manner so quickly that we are unable to detect them in our assay, these studies strongly support the conclusion that programmed DSBs are rarely formed in *vilya*[826] oocytes.

Finally, if *vilya*[826] oocytes are unable to initiate the formation of the majority of DSBs, we would predict a severe defect in the process of meiotic recombination. An analysis of meiotic recombination in two intervals that span the majority of the *X* chromosome shows a complete failure of recombination in *vilya*[826] (*Figure 2—figure supplement 2A*). Germline expression of *vilya*[3XHA] was able to fully rescue the frequency and distribution of recombination in the *vilya*[826] mutant. We also analyzed the frequency of recombination across the entire *3rd* chromosome and found that the frequency of

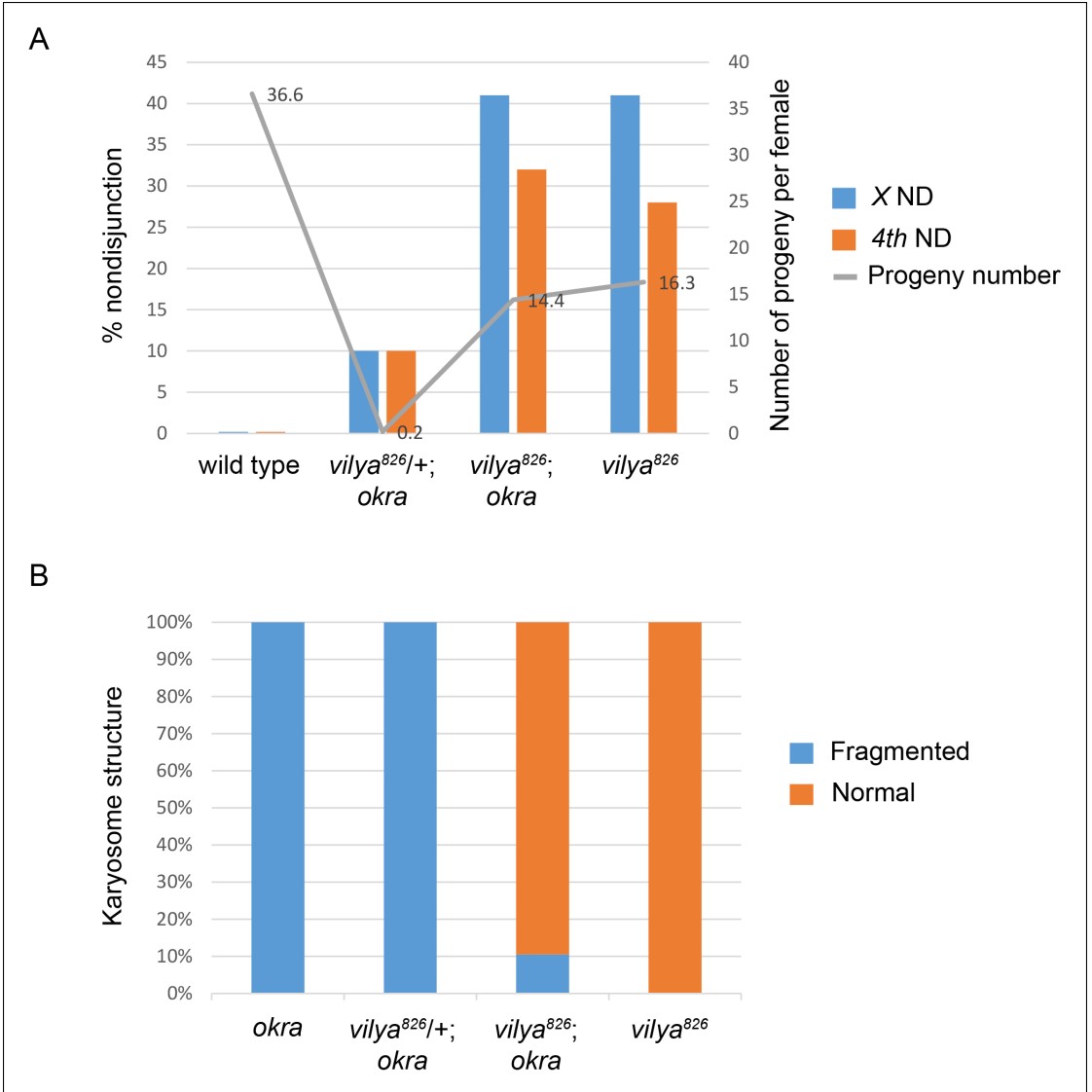

**Figure 2.** *vilya*[826] rescues the fertility and karyosome defects of the DSB repair-deficient mutant *okra*. (A) *vilya*[826] rescues the fertility defect of the DSB repair-deficient mutant *okra* and displays an increase in chromosome nondisjunction in the double mutant, similar to that of the single *vilya*[826] mutant. The fertility of *vilya*[826] is only about 30% of the wild-type control, likely due to the high levels of chromosome missegregation. The high levels of *4th* chromosome nondisjunction observed in the *vilya* mutant are due to the inability of the achiasmate segregation system to withstand the effects of a global reduction in recombination (**Zitron and Hawley, 1989**; **Hawley et al., 1992**). Average number of progeny per female (gray line) is shown. Number of adjusted progeny scored in the nondisjunction assay: wild type (330), *vilya*[826]/+; *okra* (20), *vilya*[826]; *okra* (1587) and *vilya*[826] (195). Number of females tested in the fertility assay: wild type (9), *vilya*[826]/+; *okra* (90), *vilya*[826]; *okra* (110) and *vilya*[826] (12). Wild type and *vilya*[826] data collected independently from other genotypes. ND, nondisjunction. (B) *vilya*[826] rescues the karyosome defect seen in the *okra* mutant to 89.5% of normal. Number of karyosomes analyzed: *okra* (38), *vilya*[826]/+; *okra* (20), *vilya*[826]; *okra* (12), and *vilya*[826] (6). (A,B) ( + ) indicates wild-type copy of *vilya* present on *FM7* balancer chromosome.

The following figure supplements are available for Figure 2:

**Figure supplement 1.** *vilya*[826] rescues the karyosome defect of the DSB repair-deficient mutant *okra*.

**Figure supplement 2.** *vilya*[826] is defective in meiotic recombination.

recombination was reduced over 50-fold in *vilya*[826] compared to wild type (**Figure 2—figure supplement 2B**). Taken together, the chromosome missegregation, the lack of recombination, the near absence of γH2AV staining in all nuclei in each cyst in region 2A, and the ability of *vilya*[826] to rescue the defects of a DSB repair mutant indicate that *vilya*[826] is defective in the ability to initiate programmed DSB formation.

## Vilya localizes to the central region of the SC

We analyzed the localization of Vilya throughout pachytene using the epitope-tagged germline expression construct described above that fully rescued both the nondisjunction and meiotic recombination phenotype of the *vilya*[826] mutant. The tagged Vilya construct was expressed in the female germline using the Gal4-UAS system under the control of the *nanos (nos)* promoter. Using this system, proteins are expressed throughout most stages of oogenesis at high levels (**Van Doren et al., 1998**).

Immunofluorescence analysis coupled with structured illumination microscopy (SIM) allowed us to precisely determine the localization of Vilya during pachytene. We find that during early pachytene, Vilya[3XHA] localizes to the central region of the SC in both linear stretches and discrete foci (**Figure 3A-a**). The Vilya[3XHA] linear tracks appear within the central region of the SC, as the fluorescence is seen in between the two lateral sides of the SC using an antibody that localizes to the C terminus of the TF protein C(3)G (**Anderson et al., 2005**, **Collins et al., 2014**). In addition, the discrete Vilya[3XHA] foci can also be seen within the central region (**Figure 3A-b**).

As the cyst progresses to early/mid-pachytene, the linear tracks of Vilya become less apparent and the foci become more discrete (**Figure 3A-c,d**). We counted the number of discrete foci throughout each stage of pachytene in the germarium and found that similar to the trend in γH2AV foci number (see **Figure 1C**), Vilya[3XHA] foci are most abundant in region 2A (average 8 foci, SD = 2) (the stage at which programmed DSBs are being induced) and then decline gradually throughout early/mid-pachytene (region 2B average 4.4, SD = 0.8). In mid-pachytene (region 3), a stage in which DSB repair is underway or complete, we still see an average of 3.2 Vilya[3XHA] foci (SD = 0.9) (compare **Figure 1C** and **Figure 3B**). The observation that Vilya[3XHA] persists at discrete foci after DSB repair has begun and crossovers are forming suggests that Vilya plays a role in the completion of actual crossovers, such as in the breaking and exchange of LEs.

In those mid-pachytene region 3 oocytes that lack discrete Vilya[3XHA] foci, Vilya[3XHA] is localized exclusively throughout the entire central region of the SC (**Figure 3—figure supplement 1A**). This localization pattern is also observed in late pachytene egg chambers, those that have matured past the germarium (**Figure 3—figure supplement 1A–B**). The absence of the discrete Vilya[3XHA] foci at mid-pachytene may suggest that the foci seen earlier have disassembled; however, the significance of this relocalization of Vilya[3XHA] to the central region at the later stages of pachytene is unclear. The specificity of the anti-HA antibody for both types of Vilya[3XHA] staining (discrete foci and linear tracks) can be seen in **Figure 3—figure supplement 1B** where there is a complete absence of staining on wild type tissue.

To further characterize the localization of Vilya[3XHA] foci, we analyzed the number and distribution of Vilya[3XHA] foci within the SC in early/mid-pachytene region 2B oocytes. At this stage, the Vilya[3XHA] foci are readily visible, and the SC becomes shorter and thicker than in early pachytene nuclei. Using SIM, 3D visualization, and the spot function in Imaris, we were able to trace five independent tracks of SC in five oocytes and determine the distribution of Vilya[3XHA] foci within each SC track. Examples of the traced SC in oocytes can be seen in **Figure 3C-a,b**; **Figure 3—figure supplement 1C** and **Video 1**, and linearized traces can be seen in **Figure 3C-c** and **Figure 3—figure supplement 1D**. We presume that each of the linear tracks correspond to the euchromatic SC (the well-defined SC which is visibly more structured in immunofluorescence assays than is the less-defined heterochromatic/pericentromeric SC) of the five major chromosome arms (*2L, 2R, 3L, 3R* and the *X* chromosome). We cannot discern the SC of the small *4th* chromosomes, nor can we trace through the less-distinct SC near the pericentromeric heterochromatin (**Carpenter, 1975a**) to the other chromosome arm of the same chromosome.

This analysis was designed to tell us whether the distribution of foci within the SC was consistent between oocytes, and whether or not the position of the foci within the SC of each chromosome arm might suggest a possible link to the position of crossovers (an average of one crossover per chromosome arm within the euchromatic SC). While a strong correlation between crossover position

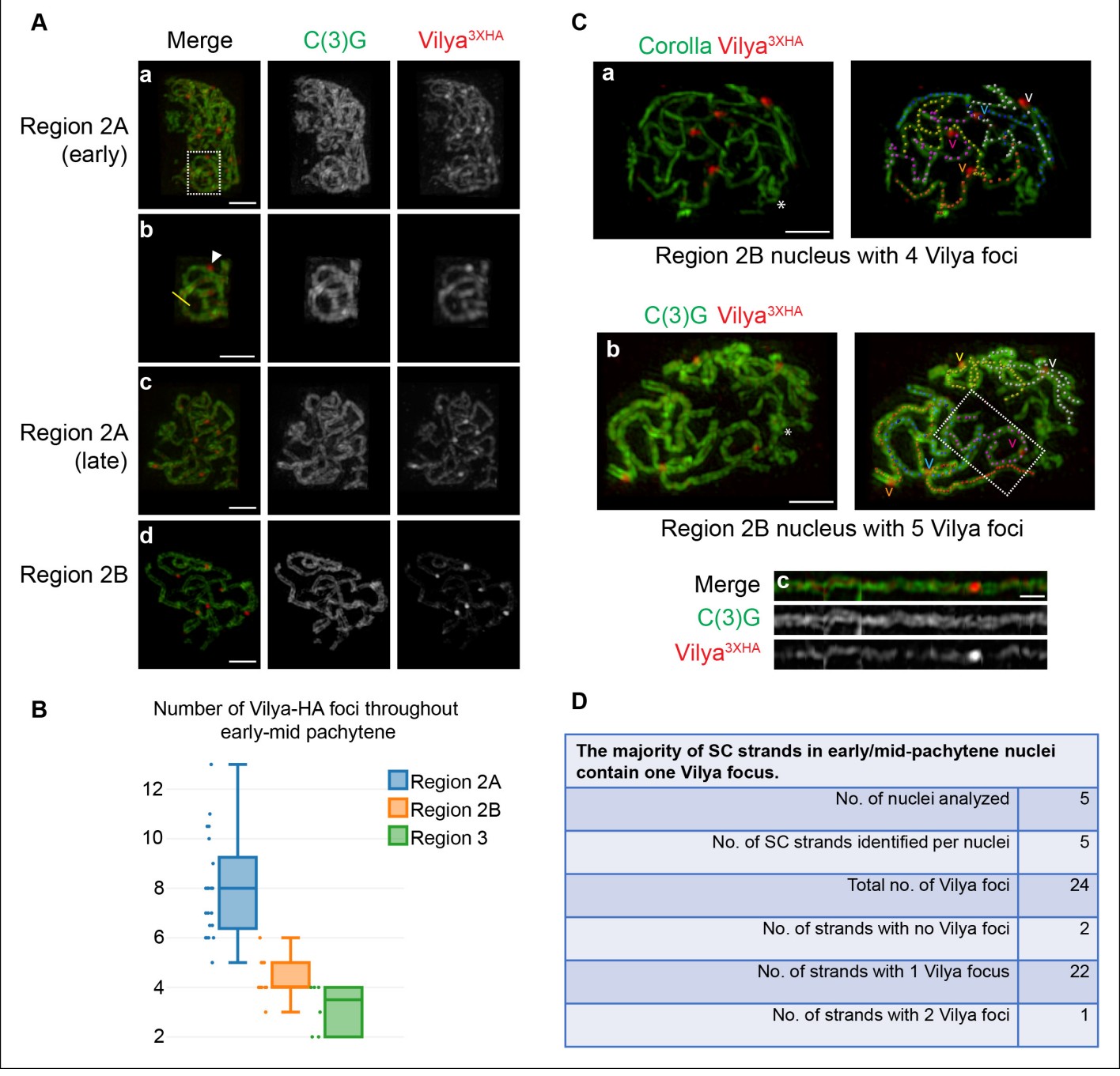

**Figure 3.** Vilya localizes to the central region of the SC in both linear elements and discrete foci. (**A**) Localization of Vilya[3XHA] throughout early pachytene as assayed by germline expression of *vilya[3XHA]* using antibodies to the transverse filament protein C(3)G (green) and an antibody to HA (red). Images are maximum intensity projections of deconvolved z-series from a DeltaVision OMX microscope through the selected nuclei. Scale bar, 1 μm. (**A-a**) Early pachytene (region 2A) oocyte nucleus showing that Vilya localizes to the central region of the SC in both linear strands and discrete foci. (**A-b**) Higher magnification of the white dashed box in A showing Vilya[3XHA] clearly positioned in the central region between the two tracks of C(3)G (yellow line) and a discrete Vilya[3XHA] focus sitting within and above a stretch of SC (arrowhead). (**A-c,d**) Localization of Vilya[3XHA] in region 2A and 2B showing the discrete foci and SC staining. Note in region 2B the SC shortens. (**B**) Analysis of the number of Vilya[3XHA] foci throughout early/mid-pachytene. (**C-a, b**) Traces of SC between homologous chromosome arms in early/mid-pachytene (region 2B) nuclei expressing *vilya[3XHA]*. Images are maximum intensity projections of deconvolved z-series from a DeltaVision OMX microscope through the selected nuclei. (*) Indicates the chromosome center containing pericentric heterochromatin and is the location of the centromeres. Scale bar, 1 μm. Individual tracks of SC between homologous chromosome arms were identified and each labeled with a separate color. The corresponding Vilya[3XHA] foci associated with each stretch of SC between homologous chromosome arms are labeled by a (v) in the same color as the stretch of SC it is on. (**C-a**) The oocyte nucleus is labeled with antibodies to Corolla

*Figure 3. continued on next page*

*Figure 3. Continued*

(green) and HA (red). This nucleus has five colored chromosome arms and four Vilya[3XHA] foci. Each chromosome arm has been linearized in *Figure 3—figure supplement 1D*. (C-b) Oocyte nucleus is labeled with antibodies to C(3)G (green) and HA (red). This nucleus has five colored chromosome arms and five Vilya[3XHA] foci. (C-c) The chromosome arm outlined with the white dashed line in (C-b) has been linearized. (D) The majority (92%) of Vilya[3XHA] foci in the five nuclei that have been identified as having five clearly identifiable chromosome arms each localize to one strand. One chromosome arm contains two foci, and two chromosome arms contain no Vilya[3XHA] foci.

The following figure supplements are available for Figure 3:

**Figure supplement 1.** Localization of Vilya[3XHA] within pachytene nuclei.

**Figure supplement 2.** Vilya3XHA foci are not found at centromeres in early/mid-pachytene.

and the distribution of Vilya foci would strongly support the hypothesis that the establishment of discrete Vilya foci plays a role in crossover formation, the finding of a lack of consistency for the distribution and/or in the position of the foci might indicate the foci are an artifact from using this overexpression system. A summary of the number and distribution of Vilya[3XHA] foci from the five early/mid-pachytene nuclei in which we could clearly identify all five SCs between homologous chromosome arms is shown in *Figure 3D*. Examining these five oocyte nuclei, which contain a total of 25 stretches of SC, we observed 24 Vilya[3XHA] foci or an average of 4.8 foci per oocyte. The majority (22/24) of the SC between homologous chromosome arms were associated with only one Vilya[3XHA] focus. A small fraction (1/22) contained two foci (corresponding nucleus shown in *Figure 3—figure supplement 1C*), and 2/22 were not associated with any Vilya[3XHA] foci. These numbers correspond well to the observed distribution of crossovers in *Drosophila melanogaster*. In addition, as is seen in the images in *Figures 3C-a,b* and corresponding *Video 1*, the Vilya[3XHA] foci were found exclusively within the euchromatic SC, and no Vilya[3XHA] foci were detected in the less-defined heterochromatic SC. Consistent with this observation we failed to detect any colocalization of Vilya[3XHA] foci with the histone variant (CID, the CENP-A homolog) that localizes in the pericentromeric heterochromatin throughout early/mid-pachytene (*Figure 3—figure supplement 2*). Thus the number of Vilya[3XHA] foci in early/mid-pachytene oocytes are consistently found and correspond well to the known number and position of crossover events in flies, with each stretch of euchromatic SC between homologous chromosome arms primarily containing one focus (*Lindsley et al., 1977*).

## Vilya localizes to RNs as detected by immuno-EM

Due to the position and distribution of Vilya[3XHA] foci on the SC between homologous chromosome arms during early/mid-pachytene, as well as the persistence of these foci into mid-pachytene, we speculated that Vilya might be localizing to sites of crossovers. Unlike many model organisms where crossover-specific proteins have been identified and reagents have been made to analyze their localization, no such proteins or reagents exist in Drosophila. However, early studies by Carpenter in Drosophila show that sites of crossovers form large electron-dense structures known as RNs within the central region of the SC (*Carpenter, 1975a*; *Carpenter, 1975b*).

We performed immuno-EM using a secondary antibody labeled with gold particles on oocytes expressing *vilya[3XHA]*. In the analysis of 50 nm sections we frequently observed gold particles localizing to electron-dense RNs.

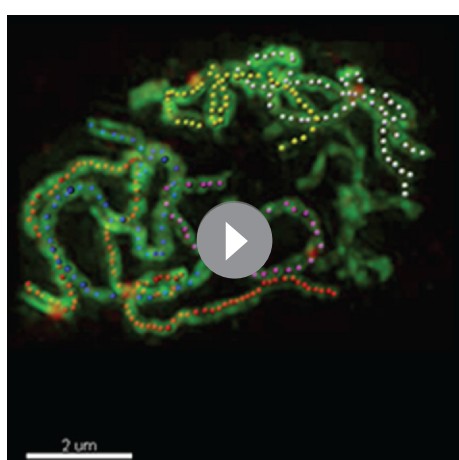

2 um

**Video 1.** Rotation of an early/mid-pachytene nuclei. Movie showing X and Y rotation of the nucleus in *Figure 3C-b*. Each chromosome arm is marked with a separate color. The SC is labeled using an antibody to C(3)G (green), and Vilya[3XHA] foci are identified with an antibody to HA (red).

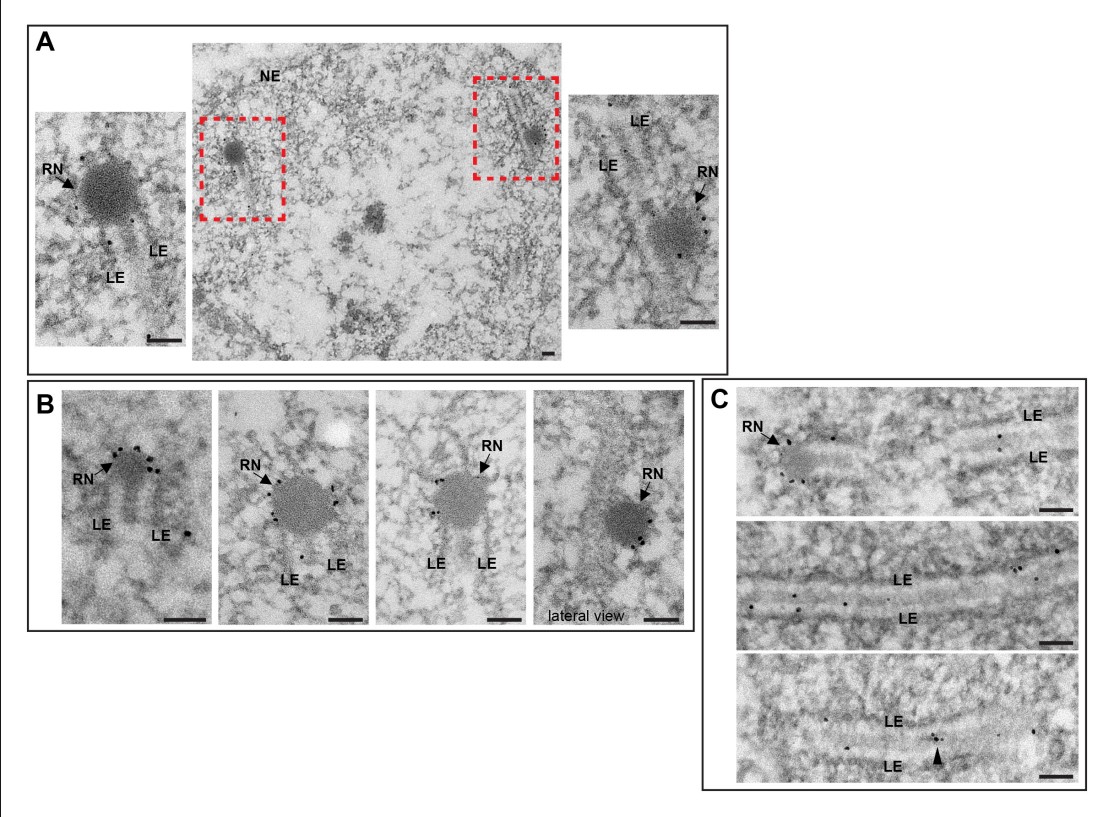

**Figure 4.** Immuno-EM of Vilya[3XHA] shows localization to both the RNs and to the central region of the SC. Immuno-gold labeling of Vilya[3XHA] from germline-expressed *PUASp-vilya[3XHA]* ovaries. (**A**) A low magnification image of a section from a single nucleus with two RNs (outlined with red dashed box). A higher magnification of each RN with associated gold particles is also shown. (**B**) Four additional immuno-EM images showing gold particles associated with RNs. A lateral view of an RN is also shown. (**C**) Three immuno-EM images showing gold particles distributed throughout the central region of the SC, as well as at RNs. Arrowheads point to cluster of gold particles in what appears to be a small electron-dense region in the central region. NE, nuclear envelope; RN, recombination nodule; LE, lateral element. Scale bar, 100 nm.

Examples are shown in *Figure 4*. In one image in *Figure 4B* we have captured what we believe to be a lateral view of a Vilya[3XHA]-associated RN sitting in and above the SC. In addition to the localization at RNs, we were able to detect gold particles throughout the entire central region of the SC, as well as small clusters of gold particles in what appear to be small electron-dense regions of the central region (*Figure 4C*). From these studies, we conclude that the discrete Vilya[3XHA] foci we detect within the central region of the SC by immunofluorescence correspond to the EM structure of the RNs, and/or their precursors, and are the sites of crossing over.

## The formation of discrete Vilya[3XHA] foci is dependent on programmed DSB formation but not the SC

Our data above indicate that Vilya is required for programmed DSB formation and localizes to the sites of crossing over, therefore we next wanted to determine whether these discrete foci were forming at sites of DSBs. We first analyzed the localization of Vilya[3XHA] in the absence of *mei-P22* or *mei-W68*, two genes whose function is absolutely required for DSB formation (*McKim and Hayashi-Hagihara, 1998*, *Liu et al., 2002*). Unlike in the presence of one wild-type copy of *mei-P22* (*Figure 5A*) or *mei-W68* (*Figure 5B*), in the homozygous mutant backgrounds (*Figure 5C,D*), Vilya[3XHA] is found exclusively and uniformly throughout the central region of the SC and fails to localize to discrete foci, indicating that DSB formation is required for the localization of Vilya[3XHA] to discrete foci but not for the linear localization to the central region of the SC.

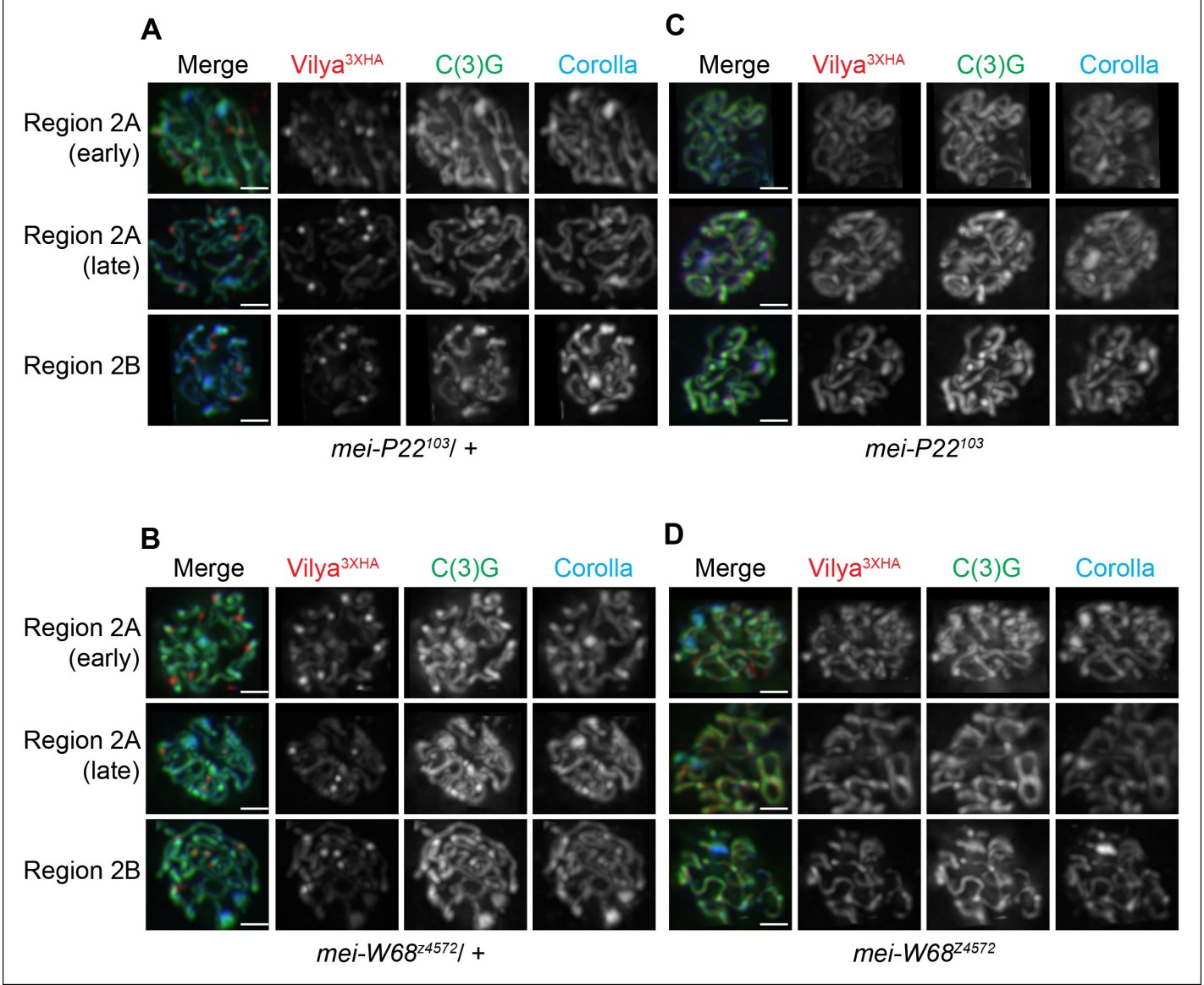

**Figure 5.** Localization of Vilya[3XHA] to discrete foci is dependent on the process of DSB formation. (A-D) Immuno-localization of Vilya[3XHA] foci in the presence and absence of DSB formation. Pachytene nuclei in the specified regions were labeled with antibodies to HA (red), C(3)G (green) and Corolla (blue). Images are maximum intensity projections of deconvolved z-series from a DeltaVision microscope through the selected nuclei. Scale bar, 1 μm. (A-B) Germline expression of *PUASp-vilya[3XHA]* in the presence of DSB formation. (**A**) *y w nos-Gal4/w; PUASp-vilya[3XHA]/+; mei-P22[103]/+.* (**B**) *y w nos-Gal4/w; PUASp-vilya[3XHA]/mei-W68[z4572].* (**C,D**) Germline expression of *PUASp-vilya[3XHA]* in the absence of DSB formation. (**C**) *y w nos-Gal4/w; PUASp-vilya[3XHA]/+; mei-P22[103].* (**D**) *y w nos-Gal4/w; PUASp-vilya[3XHA] mei-W68[z4572]/mei-W68[z4572].*

Since Vilya[3XHA] foci do not form in the absence of DSBs, we next examined whether these discrete foci are specifically forming at DSB sites. We performed immunofluorescence analysis to determine if Vilya[3XHA] foci are associated with γH2AV, the histone modification that occurs immediately following DSB formation. We find that 60.5% of the Vilya[3XHA] foci are closely associated with γH2AV (49 of the 81 Vilya[3XHA] foci from 11 early pachytene nuclei). The immunofluorescence signals for Vilya[3XHA] and γH2AV can be seen in region 2A as foci that colocalize or foci that are adjacent to, but cannot be separated from, each other in a single z-section (*Figure 6A-a'*) (see Materials and methods). As the γH2AV modification at the DSB site can spread some distance (*Rogakou et al., 1999*; *Downs et al., 2004*; *Shroff et al., 2004*), in this experiment we considered both types of localization for Vilya[3XHA] and γH2AV as being associated.

Since the process of DSB formation and repair is a dynamic one, we wanted to verify that the degree of association between Vilya[3XHA] and the γH2AV modification at DSB sites was significant. We performed randomized controls, rotating the Vilya[3XHA] image stack, to determine the degree of colocalization that would occur by chance for the 11 oocytes analyzed above (see Materials and Methods for details). In the 11 early pachytene oocytes that showed an association frequency of 60.5%, only 8.6% (7 of the 81 Vilya[3XHA] foci) remained associated after rotation of the Vilya[3XHA] channel, suggesting the observed degree of association between Vilya[3XHA foci] and the γH2AV modification cannot be explained by coincidence (p < 0.0001, binomial test).

We also analyzed the number of Vilya[3XHA] foci and their association with DSBs in a DNA repair-deficient mutant. For this experiment we chose to use the mutant *spnB* (*Ghabrial et al., 1998*), which is located on a separate chromosome from both the expression construct and the germline driver and could easily be combined genetically for this analysis. SpnB, the XRCC3 or Rad51-like protein, is required for programmed DSB repair, and therefore in the absence of *spnB* function, DSBs fail to be repaired and can be seen by immunofluorescence as γH2AV foci accumulating in mid-pachytene oocytes (region 3). We find that in the absence of DSB repair, the number of Vilya[3XHA] foci in region 3 increases from an average of 3.2 (SD = 0.9) in an otherwise wild-type background (*Figure 3B*) to 7.5 (SD = 1.6, n = 10 oocytes). The frequency of Vilya[3XHA] foci associated with γH2AV marks was 68% (51 of the 75 Vilya[3XHA] foci in 10 oocytes), comparable to the 60.5% seen in early pachytene when DSB repair is progressing normally (p = 0.93, binomial test) (*Figure 6B and 6B-b'*). In addition, similar to the DSB repair-proficient background above, in the absence of DSB repair the frequency of association between Vilya[3XHA] and γH2AV was reduced to 10.6% (8 of the 75 Vilya[3XHA] foci) upon rotation of the Vilya[3XHA] channel (p < 0.0001, binomial test).

In order to determine whether the SC was required for Vilya[3XHA] to localize properly at pachytene, we expressed *vilya[3XHA]* in the absence of the TF protein C(3)G and assayed for the presence of both linear staining and discrete Vilya[3XHA] foci. The SC is not required for the temporal induction of DSBs in nurse cells in early pachytene (see *Figure 1—figure supplement 4*), however it is required for wild-type levels of DSB formation in oocytes (*Mehrotra and McKim, 2006*). Although it is difficult to distinguish oocyte nuclei from nurse cell nuclei in the absence of *c(3)G*, occasionally the oocyte can be located by the weak haze of nuclear Corolla staining (*Collins et al., 2014*). As shown in *Figure 7*, we find that in the absence of C(3)G, Vilya[3XHA] is able to localize in early pachytene oocytes to discrete foci that are often associated with γH2AV marks (75% of Vilya[3XHA] foci, 34 of 45, colocalize with γH2AV marks in 14 oocyte nuclei analyzed). As the number of DSBs in oocytes of a *c(3)G* mutant are reduced to 15–20% of normal (*Mehrotra and McKim, 2006*), as expected, Vilya[3XHA] foci are also reduced in number compared to wild type in region 2A (an average of 3.2 Vilya[3XHA] foci per oocyte compared to 8.0 in wild type). Interestingly, we failed to see a persistence of Vilya[3XHA] in region 2B oocytes that we could identify by Corolla staining. Of the

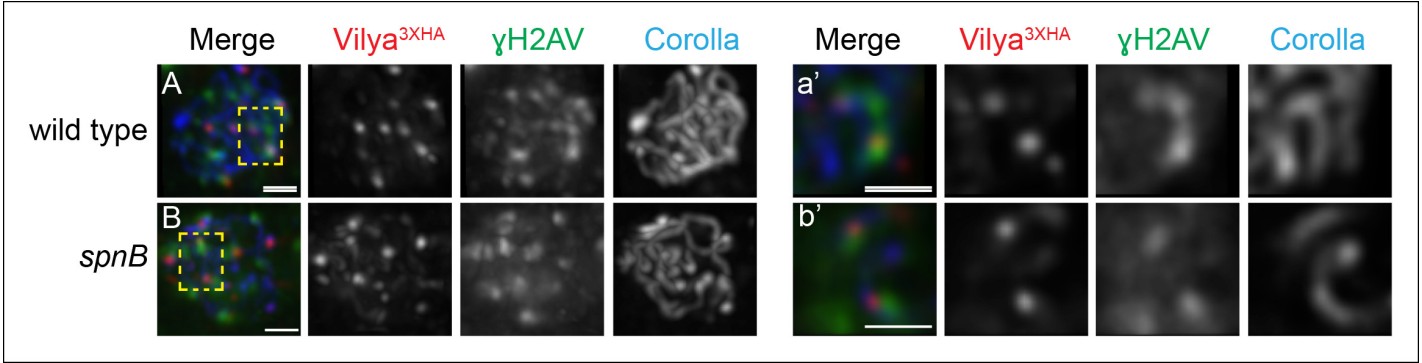

**Figure 6.** A subset of Vilya[3XHA] foci localize near γH2AV staining at DSB sites. (**A,B**) Immunofluorescence analysis of Vilya[3XHA] (red) localization at the sites of programmed DSBs that are recognized with the γH2AV modification (green) and Corolla (blue) in region 2A (**A**) or region 3 (**B**) pachytene nuclei. Images are maximum intensity projections of deconvolved z-series from a DeltaVision microscope through the selected nuclei. (A-a', B-b',) Higher magnification of a single z-section from a small region outlined in the yellow box of the corresponding image A-B, respectively, showing the close association of Vilya[3XHA] with γH2AV marks. Genotype for (**A**) wild type in this figure refers to the genotype *y w nos-Gal4/+; PUASp-vilya[3XHA]/+; mei-P22[103]/+. mei-P22* is not haploinsufficient. (**B**) *y w nos-Gal4/+; PUASp-vilya[3XHA]/+; spnB*. Scale bar, 1 µm.

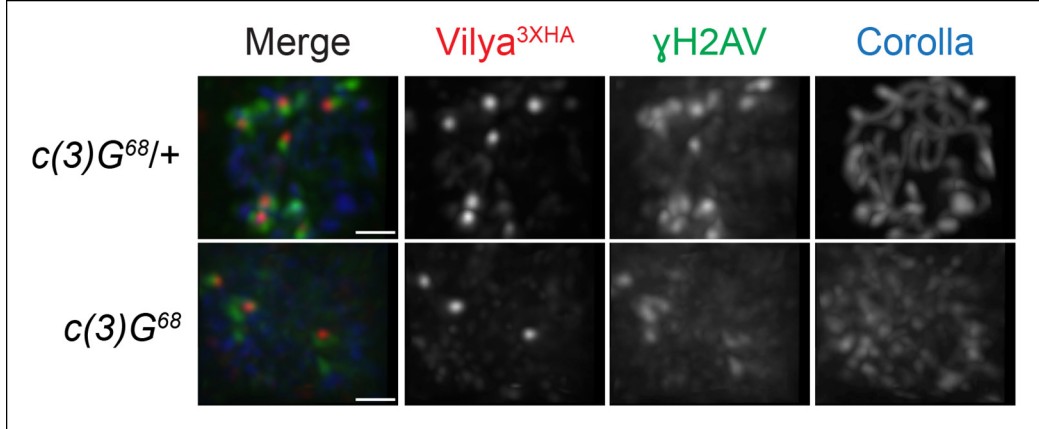

**Figure 7.** Localization of Vilya³ˣᴴᴬ to discrete foci in early pachytene is not dependent on the SC. Immunofluorescence analysis of *vilya³ˣᴴᴬ*-expressing region 2A oocytes in the presence and absence of *c(3)G*. Early pachytene nuclei in region 2A were labeled with antibodies to HA (red), γH2AV (green) and Corolla (blue). Images are maximum intensity projections of deconvolved z-series from a DeltaVision microscope through the selected nuclei. Scale bar, 1 μm.

seven region 2B oocytes we analyzed that no longer contained γH2AV staining, only two had a single Vilya³ˣᴴᴬ focus while the remaining had none. We speculate that the absence of Vilya³ˣᴴᴬ foci at this stage is a consequence of a failure to form RNs and thus repair those DSBs into crossovers. At this time, however, we cannot rule out the possibility that the levels of expression of *vilya³ˣᴴᴬ* using the Gal4-UAS system in the *c(3)G* mutant is less than in our wild-type background. In addition, we never observed linear Vilya³ˣᴴᴬ staining in the absence of *c(3)G*, indicating that the SC is required to localize Vilya to the central element but not to DSBs.

These results strongly support the view that Vilya plays a crucial role in DSB formation, and its localization to discrete foci requires genes whose products are known to be involved in the induction of DSBs. In the absence of DSB formation, localization of Vilya³ˣᴴᴬ to the central region in linear tracks, which is dependent on the SC, is not disrupted; however discrete Vilya³ˣᴴᴬ foci do not form. Interestingly, in the absence of DSB repair, the number of discrete Vilya foci increases at unrepaired DSB sites in mid-pachytene. It is currently unclear as to whether this increase in Vilya³ˣᴴᴬ foci in the absence of DSB repair indicates more DSBs are selected to become crossovers, or whether we may be underestimating the number of Vilya foci throughout pachytene due to the temporal nature of DSB formation and repair.

## Exogenous DSBs can recruit Vilya³ˣᴴᴬ to discrete foci

To determine whether exogenous DSBs can recruit Vilya to them, which would support a downstream role for Vilya in addition to its role in DSB formation, we analyzed whether the localization pattern of Vilya³ˣᴴᴬ in a *mei-W68* mutant changed after X-ray treatment. We speculated that if Vilya only plays a role in DSB formation, exogenous DSBs would fail to recruit Vilya³ˣᴴᴬ. However, if Vilya was also required for a downstream function at the RNs, these lesions may indeed recruit and concentrate Vilya³ˣᴴᴬ to them, and thus we would observe the exclusive linear staining change in this mutant background after X-ray treatment.

In the presence of germline expressed *PUASp-vilya³ˣᴴᴬ* in a *mei-W68* mutant, where discrete Vilya³ˣᴴᴬ foci are not detected (see **Figure 5D** and **Figure 8**), we exposed females to X-rays and looked for the appearance of Vilya³ˣᴴᴬ foci after 5 hr, a timeframe previously shown to have γH2AV signal present at X-ray-induced lesions throughout early/mid-pachytene oocytes (**Mehrotra and McKim, 2006**). We find that in some instances, Vilya³ˣᴴᴬ foci could be detected at γH2AV marks (**Figure 8**). These Vilya³ˣᴴᴬ foci appear to be discrete foci and are different from the often observed, more concentrated areas of linear Vilya³ˣᴴᴬ staining associated with dense regions of SC (based on Corolla staining) in the *mei-W68* mutant background in the absence of X-ray (**Figure 8**). However, in the X-ray-treated females, we also observed instances of Vilya³ˣᴴᴬ foci that were not associated with

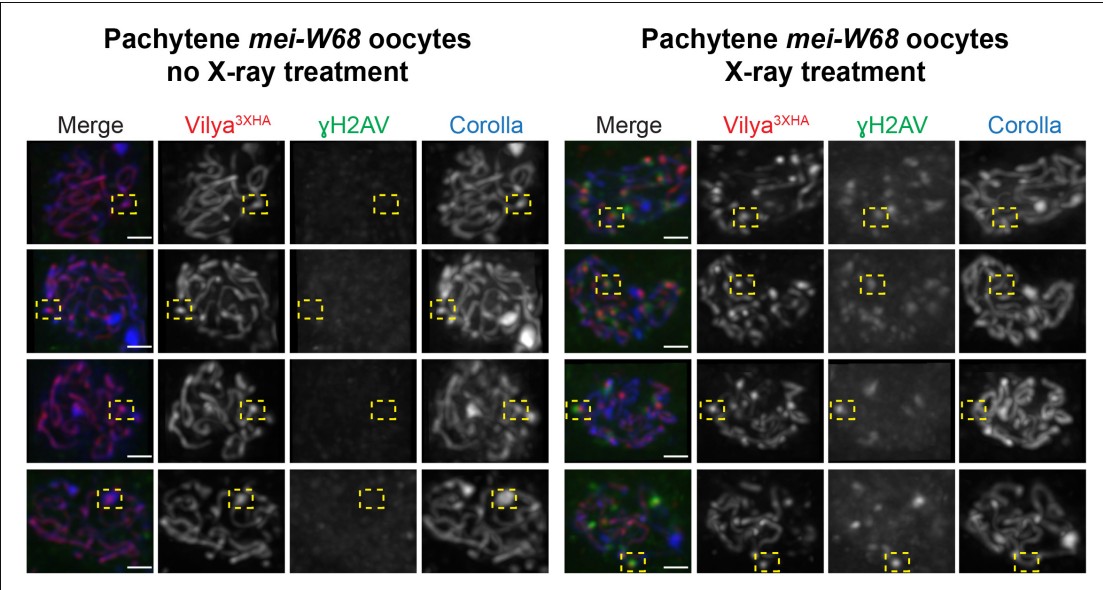

**Figure 8.** Some DSBs created by X-ray can recruit Vilya3XHA to discrete foci. Immunofluorescence analysis of germline expression of *PUASp- vilya*[3XHA] in the absence of functional *mei-W68* with and without X-ray treatment. Ovaries were stained with antibodies to HA (red), γH2AV (green) and Corolla (blue). Four examples of early/mid-pachytene oocytes are shown for each treatment. Yellow boxes in the no X-ray treatment show examples of concentrated regions of linear Vilya[3XHA] that are associated with dense Corolla staining. Yellow boxes in the X-ray treatment show examples of discrete Vilya[3XHA] foci associated with γH2AV marks that are not associated with dense Corolla staining. Scale bar, 1 μm.

γH2AV signals and many γH2AV marks not associated with Vilya[3XHA] foci. We should also note that X-ray-induced DSBs did not appear to significantly alter the localization of linear tracks of Vilya[3XHA] in this mutant background, in that we did not see significant removal of Vilya[3XHA] from the central region of the SC in oocytes with high levels of X-ray-induced breaks.

Although we do not know at this time whether the X-ray-induced DSBs that recruit Vilya[3XHA] to them can be processed into crossovers, the ability of X-ray-induced lesions to concentrate and form discrete Vilya[3XHA] foci at some of them suggests that Vilya is playing an active role in the processing of DSBs.

## Vilya physically interacts with Mei-P22

Mei-P22 is a relatively small protein that localizes to discrete foci prior to DSB formation, is required for DSB formation, and partially colocalizes with γH2AV (*Liu et al., 2002*, *Mehrotra and McKim, 2006*). Because of this, we wondered whether Vilya, which is also required for DSB formation, and Mei-P22 directly interact. In the yeast two-hybrid system, we found that Vilya and Mei-P22 strongly interact as assayed on the reporter plate (*Figure 9*). We determined that the Vilya[826] form of the protein could also interact with Mei-P22 in this assay, although this interaction appeared to be weaker than with full-length Vilya when controlled for plating. We also tested whether a mutant form of Mei-P22, Mei-P22[103], a nonsense mutation resulting in a premature stop codon truncating the Mei-P22 protein by 32 amino acids (*Liu et al., 2002*), can interact with Vilya. We found that this mutation, which abolishes Mei-P22 function and DSB formation in vivo, is still able to bind to both Vilya, and Vilya[826], albeit to a lesser extent for Vilya[826] when controlling for plating.

We then determined whether the RING domain of Vilya is required for this interaction by generating a series of mutations in Vilya that disrupt critical residues in the RING domain and testing each for the ability to interact with Mei-P22 in a yeast two-hybrid system. We substituted each of the cysteines for serines and the histidine for a tyrosine in the RING domain (see *Figure 1—figure supplement 1*). Each of the mutations ablates the ability of Vilya to interact with Mei-P22 (*Figure 9—figure supplement 1A*). The failure of the RING domain mutants to interact with Mei-P22 is not due to

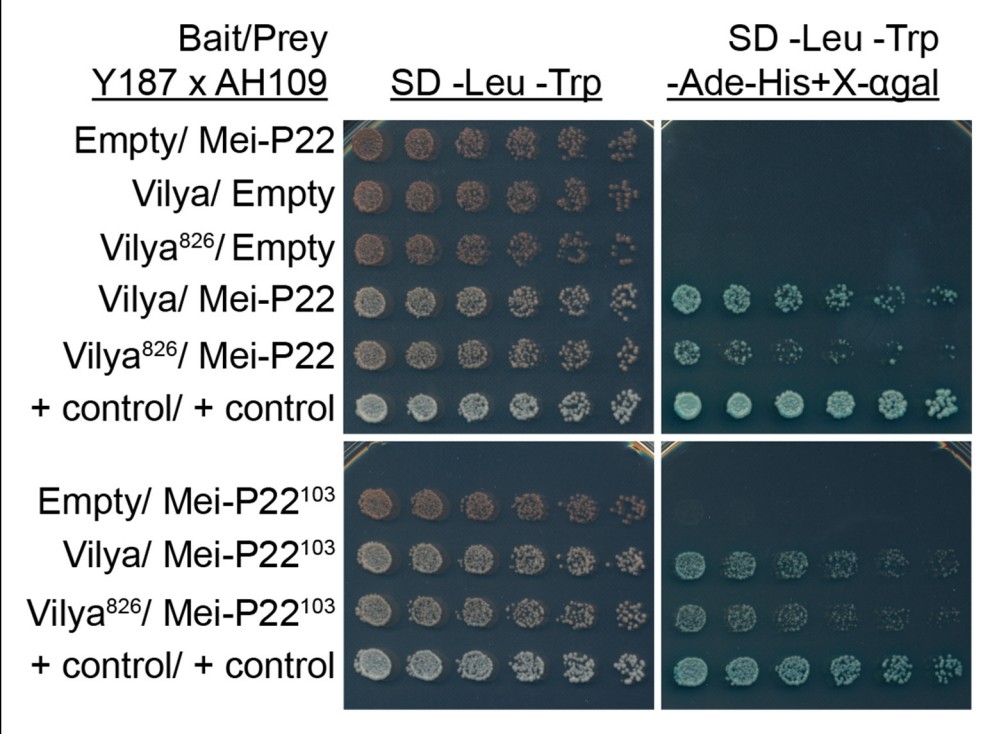

**Figure 9.** Vilya and Mei-P22 interact by yeast two-hybrid. Yeast two-hybrid was used to test for an interaction between Vilya and Mei-P22. All diploid strains, where the OD$^{600}$ was equalized before plating, grow equally well under selection for both the bait and prey plasmids (SD -Leu-Trp). Six two-fold dilutions for each diploid were plated on each selection plate. Vilya and Mei-P22 strongly interact on the reporter plate (SD -Leu-Trp-Ade-His + X-αgal). Vilya$^{826}$ and Mei-P22 also interact, but it appears to be a weaker interaction than with full-length Vilya on the reporter plate. Vilya and Mei-P22$^{103}$ interact, as well as Vilya$^{826}$ and Mei-P22$^{103}$, although this interaction is also weaker than with full-length Vilya. No interaction was detected with empty construct for any of the plasmids used. The control plasmids are pGBKT7-53 and pGADT7-T supplied by Clontech.

The following figure supplements are available for Figure 9:

**Figure supplement 1.** A functional RING domain is required for Vilya to interact with Mei-P22 in yeast-two hybrid assay.

altered protein expression levels or degradation, as we do not observe any obvious differences in the amount or size of the RING domain mutant proteins compared to wild type (*Figure 9—figure supplement 1B*). However, as is the case for virtually all yeast two-hybrid studies, we cannot fully rule out the possibility that mutating key residues of the RING domain may alter the protein conformation, thus creating a failed interaction. These studies, however, do suggest that a functional RING domain within the Vilya protein is critical for Vilya and Mei-P22 interaction.

Based on these studies, Vilya and Mei-P22 likely interact in vivo, and their interaction is dependent on the RING domain of Vilya. Previous studies using the expression construct *hsp83:mei-P22$^{3XHA}$* have shown that Mei-P22$^{3XHA}$ localizes to discrete foci, which are found on chromatin closely associated with the SC and are not dependent on DSB formation (*Liu et al., 2002*). Therefore, we anticipate that Mei-P22 localization is also not dependent on Vilya. The persistence of Vilya$^{3XHA}$ at discrete foci into early/mid-pachytene, and the absence of Mei-P22$^{3XHA}$ staining at this stage (*Liu et al., 2002*; *Mehrotra and McKim, 2006*), suggests that Vilya may have additional functions at the DSB site, such as in crossing over, that are independent from Mei-P22.

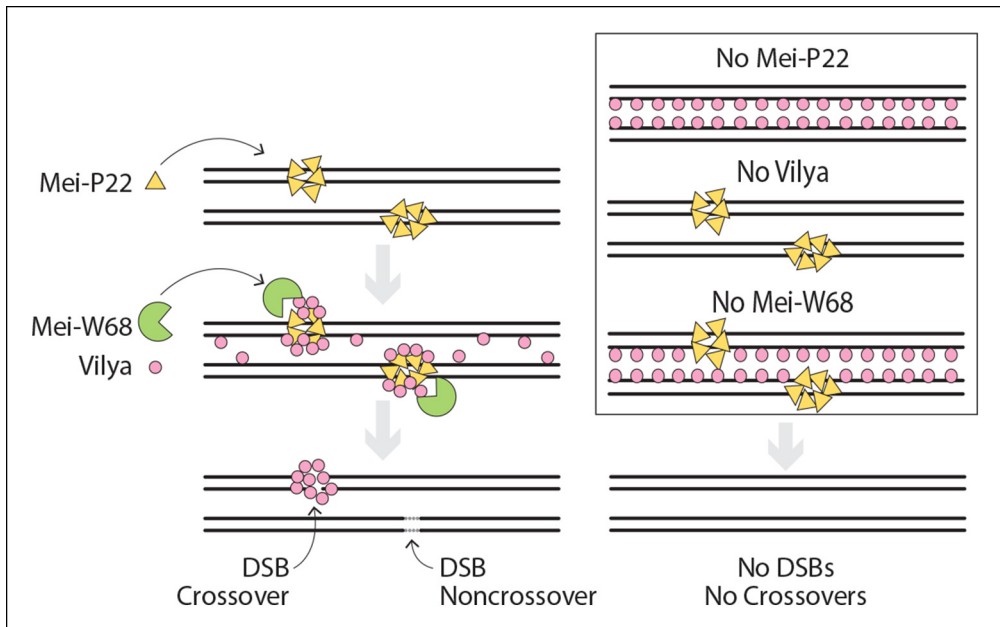

**Figure 10.** Model of DSB formation in Drosophila female oocytes. (Left) In wild-type oocytes, Mei-P22, which localizes to discrete foci prior to the time γH2AV foci are present, is located at chromatin adjacent to the SC (*Liu et al., 2002*). Vilya localizes to the central region of the SC and is required along with its binding partner, Mei-P22, and Mei-W68 (the Spo11 homolog) for formation of DSBs. Although initially Vilya may localize to a vast majority of, if not all, DSBs, as the oocytes mature into early/mid-pachytene, Vilya is retained and/or recruited to form discrete foci at sites of crossing over. Failure to accumulate Vilya at DSB sites would direct that DSB to a noncrossover fate. (Right) In the absence of Mei-P22 or Mei-W68, and thus in the absence of DSB formation, Vilya fails to localize to discrete foci and is found exclusively along the central region of the SC. In the absence of Vilya, we speculate, based on the fact that Mei-P22 can localize to discrete foci in the absence of Mei-W68 (*Liu et al., 2002*), that Mei-P22's localization is unaffected. However, DSB formation would fail due to the absence of *vilya* function. In the absence of Mei-W68, Mei-P22 is able to bind normally (*Liu et al., 2002*), however, due to the absence of DSBs, Vilya does not form discrete foci. In all these instances, crossovers do not form.

The following figure supplements are available for Figure 10:

**Figure supplement 1.** Protein alignment of Vilya to Zip3 and HEI10 homologous proteins.

## Discussion

### Vilya's role in DSB formation and crossing over

Vilya, in conjunction with another DSB accessory protein, Mei-P22, acts to facilitate the initiation of recombination during meiotic prophase. As shown in our model (*Figure 10*), DSBs are not formed in the absence of Mei-P22, Vilya, or Mei-W68 (*Dm* Spo11), resulting in the absence of crossovers (*Liu et al., 2002*, *Mehrotra and McKim, 2006*) (this study). Unlike Mei-P22, whose localization to discrete foci in early pachytene is not dependent on DSBs (or at least *mei-W68* function) but is dependent on the SC (*Liu et al., 2002*), Vilya's ability to localize to discrete foci appears to require only the formation of DSBs but not the SC. In the absence of either Mei-P22 or Mei-W68, the discrete Vilya[3XHA] foci, which are first apparent in early pachytene and often persist throughout mid-pachytene, do not form. Our studies also suggest that the localization of Vilya[3XHA] along the central region of the SC is not required for its localization to discrete foci, as we did not detect any linear staining of Vilya in a *c(3)G* mutant, but we did detect discrete foci often colocalizing with γH2AV marks (see *Figure 7*). Therefore, Vilya is not only required for the formation of DSBs, but it's localization to discrete foci, which can be found at a significant number of DSBs (based on γH2AV staining), is dependent on DSB formation. Based on these observations, along with the finding that Mei-P22 and Vilya interact in a yeast two-hybrid assay, we propose that Mei-P22 acts upstream of Vilya, and

recruits Vilya and Mei-W68, which catalyzes DSBs. Vilya is recruited to at least a subset of DSBs, which are then visualized as discrete foci in early pachytene often colocalizing with γH2AV marks.

Our model further suggests that only those DSBs that recruit sufficient Vilya to form foci are converted into crossovers and can be visualized by the discrete prominent foci seen in early/mid pachytene. We base this proposal on the observation in budding yeast that the differential enrichment of Zip3 at DSBs positively biases the DSB toward the crossover pathway (*Serrentino et al., 2013*). We know from our studies that exogenous DSBs have the ability to recruit or concentrate Vilya to a subset of them, in the absence of SC there appears to be a failure to maintain Vilya[3XHA] at discrete foci in early/mid-pachytene oocytes, and our immuno-EM analysis of oocytes expressing *vilya[3XHA]* indicate that Vilya is indeed a component of the RN. Together, these data suggest that Vilya plays an active role at the sites of crossing over. Although we do not have direct evidence that Vilya controls DSB fate by promoting crossover maturation, our demonstration that Vilya is recruited to the sites of DSBs in a similar number and position to that of RNs strongly supports this hypothesis.

## Is Vilya a Zip3 homolog?

It is very tempting to place Vilya within the Zip3 family of homologs based on findings presented here. DIOPT, the Drosophila RNAi Screening Center Integrative Ortholog Prediction Tools (*Hu, et al., 2011*), predicts that Vilya is orthologous to *S. cerevisiae* Zip3 (also known as Cst9) and *C. elegans* Zhp-3, however, no homologs for Vilya were identified in other organisms including mice and humans using this tool (FlyBase). Here, we show in *Figure 10—figure supplement 1A* the sequence alignment of Vilya to Zip3 homologs of selected plants, fungi, worms, and vertebrates. These selected sequences were used to generate a maximum likelihood tree to show the relationship of Vilya to these homologs (*Figure 10—figure supplement 1B*). Consistent with the BLAST similarity network findings of Chelysheva et al. (*Chelysheva et al., 2012*), two groups could be identified. One contains the Hei10-like homologs of plants, fungi and vertebrates; and one contains the Zip3-like members including Zip3, Zhp-3 and RNF212. This analysis suggests that evolutionarily Vilya is more closely related to the Zip3-like members than Hei10 members. In addition, based on predicted protein structure and domains, there is an overall similarity between members within the Zip3 group members and Vilya. These similarities include an N-terminal RING domain (see *Figure 10—figure supplement 1A*), which predicts E3 ligase activity, an internal coiled-coil domain, and a C-terminal serine-rich domain, which many Zip3 members possess (*Reynolds et al., 2013*). We have shown above that Vilya's RING domain is required for the interaction with Mei-P22 in a yeast two-hybrid assay, and its coiled-coil domain suggests Vilya would localize to the central region of the SC.

In addition to these three domains, Vilya also has a putative SUMO-interacting motif (SUMO-IM) and three putative RXL cyclin-binding domains that are also common to Zip3 family members (*Figure 1—figure supplement 1E*) (*Cheng et al., 2006*; *Ward et al., 2007*; *De Muyt et al., 2014*). In budding yeast, the SUMO-IM in Zip3 has been shown to be required, along with the RING domain, for its interaction with the E2 enzyme Ubc9. These domains are also thought to be required for Zip3's SUMO E3 ligase activity in vivo (*Cheng et al., 2006*). Studies in Sordaria have shown that the RXL motif of Hei10, along with the RING domain, are required to modulate the levels of SUMOylation along the SC (*De Muyt et al., 2014*). Although we have not specifically determined whether the SUMO-IM and RXL motifs in Vilya are required for its function, in preliminary studies we have not been able to detect an interaction between Vilya and the Drosophila Ubc9 in a yeast two-hybrid assay. Nor have we been able to identify SUMOylation of proteins at the SC in Drosophila oocytes using an antibody to Drosophila SUMO (Abgent AP1287b, San Diego, CA). Therefore, although structurally Vilya bears a strong resemblance to the Zip3 family, we have no evidence Vilya possess SUMO E3 ligase activity, interacts directly with an E2 enzyme, or that SUMOylation plays a role at the SC in Drosophila at this time.

Vilya is also similar to some members of the Zip3 family in terms of its dynamic localization pattern. In early pachytene, using the expression system shown in this manuscript, Vilya[3XHA] localizes along linear tracks as well as discrete foci. As the oocyte progresses into early/mid pachytene, the discrete foci predominate and the signal along the central region of the SC appears to diminish (see *Figure 3A*). In oocytes that have progressed past mid-pachytene, the discrete foci disappear and the signal along the central region becomes uniform (see *Figure 3—figure supplement 1A–B*). This dynamic localization is unlikely to be a result of the overexpression system we are using because in

this same system in the absence of meiotic recombination initiation we fail to observe this pattern and only linear central region staining is present. Moreover, similar dynamic localization patterns have been seen for *C. elegans* Zhp-3, mouse RNF212, and Arabidopsis and rice Hei10 (*Jantsch et al., 2004*; *Bhalla et al., 2008*; *Chelysheva et al., 2012*; *Wang et al., 2012*; *Reynolds et al., 2013*). The general dynamic trend seems to begin with the localization along the SC as continuous linear tracks or arrays of discrete foci, and culminates with localization at discrete foci that mark the sites of crossing over. In the absence of meiotic recombination initiation in *C. elegans* (*Bhalla et al., 2008*) and mice (*Reynolds et al., 2013*), Zhp-3 and RNF212, respectively, were also found to localize along the SC, albeit the SC of nonhomologous chromosomes in mice, and failed to coalesce into discrete foci. However, we cannot rule out that the reappearance of Vilya$^{3XHA}$ to the central region of the SC in late pachytene is not a consequence of this ectopic expression since we have not discerned a later function for Vilya.

Although Vilya clearly has a unique function not found in any other Zip3 family member so far (i.e. being required for meiotic DSB formation), other unique functions for some Zip3 family members have been identified as well. For example, Zip3 in budding yeast appears to be the only member required for SC assembly (*Agarwal and Roeder, 2000*; *Bhalla et al., 2008*), and Hei10 in Sordaria is uniquely required for processes involving spindle pole body dynamics (*De Muyt et al., 2014*). Because DSB accessory proteins are highly divergent and Drosophila appear to lack homologs of the meiosis-specific MutS complex (*Yokoo et al., 2012*), which are required to stabilize strand invasion during crossing over, and the crossover-specific complex MutLγ, to which Zip3 family members bind, perhaps then it is not surprising that Drosophila has found an unique way to couple the events of DSB formation to those of crossing over.

## Materials and methods

### Drosophila genetics

All Drosophila strains were maintained on standard food at 25°C unless otherwise noted. Descriptions of genetic markers and chromosomes can be found at http://www.flybase.org. Wild-type strains used in the manuscript were *y w FRT19A*, the parent chromosome used to generate *vilya$^{826}$*, or Canton-S. Deficiency strains used for mapping were obtained from the Bloomington Drosophila Stock Center. Deficiency *Df (1)ED6630* (BL8948) uncovers *vilya*.

Other stocks used in this study include *Pnos-Gal4::VP16* (*Van Doren et al., 1998*), *Pw +; UASp-vilya$^{3XHA}$* (this study), *net dp ho b mei-W68$^{4572}$* (*Lake, et al., 2013*) *mei-W68$^{4572}$* (*Bhagat et al., 2004*), *st spnB$^{BU}$ sr e/TM6B* (*Ghabrial et al., 1998*), *okra$^{AA}$ cn bw/CyO* and *okra$^{RU}$ cn bw/CyO* (*Ghabrial et al., 1998*), *mei-P22$^{103}$ st/TM3, Sb* and *mei-P22$^{103}$ th st cu e ca/TM3, ry Sb* (*Liu et al., 2002*) and *c(3)G$^{68}$ e* and *c(3)G$^{68}$ e ca* (*Jeffress et al., 2007*). *okra* refers to the genotype *okra$^{RU}$/okra$^{AA}$*. *mei-W68* refers to the genotype *net dp ho b mei-W68$^{4572}$/mei-W68$^{4572}$*. *mei-P22* refers to the genotype *mei-P22$^{103}$ st/mei-P22$^{103}$ th st cu e ca*. *c(3)G* refers to the genotype *c(3)G$^{68}$ e/c(3)G$^{68}$ e ca*.

### Mapping of *vilya*

The meiotic mutation *mei-826* (*Collins et al., 2012*) was mapped by standard genetic assays. Recombination mapping placed the lesion between *sc* (1A8) and *w* (3B6), and deficiency mapping placed the mutation within the interval 3B1–3C5 due to the failure of *mei-826* to complement the Bloomington deficiency *Df (I)ED6630* (BL8948). Several genes within this region were selected as potential candidates, PCR amplified, and sequenced to look for potential EMS-induced lesions. One nonsense mutation (C640T) was identified in gene *CG2709* that resulted in a stop codon at amino acid 214 (R214STOP). *CG2709* was renamed *vilya*, and the *mei-826* mutant was subsequently renamed *vilya$^{826}$*. We named this gene *vilya*, as *vilya* encodes a protein with a RING domain, and Vilya is considered to be the mightiest of the Three Rings of Power given by the Elves of Eregion.

### Transgenic rescue constructs

To obtain the coding sequence (CDS) of *vilya*, cDNA was made using ImProm-II Reverse Transcription Kit System (Promega, Madison, WI) and Trizol extracted RNA from *y w FRT19A* egg chambers (stage 1–10) with oligo-dT primer supplied with the kit. PCR was performed on the cDNA with gene-

specific primers (5'-taccatggcgaaatcacaagcagg-3' and 5'-atcgctagctcacagatcgaacga-3'), directly cloned into TOPO-TA vector (Life Technologies, Grand Island, NY), and confirmed by Sanger sequencing, resulting in the plasmid *pTOPO-vilya*. *vilya* was amplified from *pTOPO-vilya* and cloned into *pBS-KS +* (Clontech, Mountain View, CA) using primers containing *Xba*I restriction sites on both 5' and 3' ends and an internal *Nhe*I engineered restriction site immediately upstream of the stop codon (5'–ggcgtctagaatggcgaaatcacaagcaggtc-3' and 5'-ctggtctagatcagctagccagatcgaac-gagttgttcggc-3'). The *Nhe*I site was used to clone in the 3X hemagglutinin (3XHA) tag that had been previously amplified from the *pPFHW* vector (DGRC, Bloomington, IN) with primers containing flanking *Nhe*I sites (5'-tcgcgctagctacccatacgatgttcc-3' and 5'-gctcgctagcagcgtaatctggaacg-3') to create the vector *pBS-vilya*$^{3XHA}$. After confirmation of sequence and orientation of 3XHA, *vilya*$^{3XHA}$ was digested out of *pBS-vilya*$^{3XHA}$ with *Xba*I and cloned into *pUASp-attB* (*Takeo et al., 2010*) at the *Xba*I site and sequenced for directionality. *pUASp-attB-vilya*$^{3XHA}$ was introduced into Drosophila by targeted integration using the *attP-40* line (Genetic Services, Boston, MA).

## Meiotic nondisjunction and recombination assays

To measure the frequency of nondisjunction and meiotic recombination on the *X* chromosome, virgin females of the listed genotype were crossed individually to *y sc cv v f / B[S]Y* males (*Zimmering, 1976*; *Matsubayashi and Yamamoto, 2003*). To assay meiotic recombination, only female progeny resulting from the above cross were analyzed for the markers *cv, v* and *f*. To obtain the frequency of nondisjunction when using *Df (1)ED6630,* normal male progeny were doubled due to the inability to recover *Df (1)ED6630* males in the assay.

To assay both *X* and *4th* chromosome nondisjunction, tester female virgins were crossed to *XY, In (1)EN,v f B; C(4)RM,ci ey*$^R$ males. Calculations were performed as previously described (*Zitron and Hawley, 1989*; *Hawley et al., 1992*).

To assay meiotic recombination on the 3rd chromosome, tester female virgins (+/+; *ru h th st cu sr e ca*/+ or *vilya*$^{826}$; *ru h th st cu sr e ca*/+) were crossed to +/Y; *ru h th st cu sr e ca* males and the resulting female progeny were scored for all markers.

To verify *vilya* was not haploinsufficient, we assayed as above for a meiotic defect of the *X* chromosome deficiency (BL8948) with the original *y w FRT19A* chromosome that the mutation was induced on (*Collins et al., 2012*). No meiotic phenotype was observed, indicating that *vilya* is not haploinsufficient (data not shown).

## Yeast two-hybrid

The Matchmaker Gold Yeast Two-Hybrid System User Manual (Clontech, Mountain View, CA) was followed for yeast transformation and for yeast two-hybrid assays. AH109 yeast were used in place of Y2Hgold. AH109 genotype is as follows: *MATa, trp1-901, leu2-3, 112, ura3-52, his3-200, gal4Δ, gal80Δ, LYS2: : GAL1$_{UAS}$-GAL1$_{TATA}$-HIS3, GAL2$_{UAS}$-GAL2$_{TATA}$-ADE2, URA3: : MEL1$_{UAS}$-MEL1 $_{TATA}$-lacZ*. Y187 genotype is as follows: *MATα, ura3-52, his3-200, ade2-101, trp1-901, leu2-3, 112, gal4Δ, met–, gal80Δ, URA3: : GAL1$_{UAS}$-GAL1$_{TATA}$-lacZ*. Bait and prey vectors used were *pGBKT7* and *pGADT7*, and cDNAs were cloned into the vectors using compatible restriction sites within the vector and contained within the primers. The CDS for *vilya* was obtained as above using primers 5'-gcggcatatggcgaaatcacaagcaggtc-3' and 5'-tcgcctgcagtcacagatcgaacgagttg-3' for full-length *vilya* or 5'-gcggcatatggcgaaatcacaagcaggtc-3' and 5'-tcgcctgcagtcagcgtcgactggaggac-3' for *vilya*$^{826}$. The CDS for *mei-P22* was obtained from the DNA of Canton-S and is identical to the sequence on Fly-Base (D. melanogaster Release 6). Primers used for cloning full-length *mei-P22* were 5'-ggcgtcgca-tatggacaggacaacagttgt-3' and 5'-ggcgctcgagctaaggtacttccaattc-3', and primers used for cloning *mei-P22*$^{103}$ were 5'-ggcgtcgcatatggacaggacaacagttgt-3' and 5'-ggcgctcgagtcactccaagtcaacgttcaa-catgg-3'.

Mutations in *vilya* cDNA were made using the Quik Change II XL Site-Directed Mutagenesis Kit (Stratagene, CA). The vector *pTOPO-vilya* (above) was used to generate the site-directed mutants, and each was cloned into the yeast expression vector (pGBKT7) using the primers above. Protein expression of each of the Vilya point mutants, as well as full-length wild-type protein, was verified by Western blotting. Briefly, a 50 mL culture of transformed Y187 yeast cells was inoculated from an over-day 5 mL culture in minimal media lacking Trp. After 7 hr, the OD$_{600}$ was determined, and equal amounts of yeast based on OD$_{600}$ were pelleted before being frozen. Cells were thawed into

0.1 M sodium hydroxide containing 1X protease inhibitor (Sigma, MO) and incubated at RT for 5 min. Equal amounts of 2X SDS loading dye containing β-mercaptoethanol was added, samples were boiled 5 min, pelleted, and lysate loaded onto a 12% SDS-PAGE gel. Protein was transferred onto PVDF membrane. For detection of expressed protein, an antibody to the c-Myc epitope, found within the pGBKT7 vector upstream and in-frame, was used (anti-c-Myc clone 9E10, Abcam, MA) at 1:1000 dilution overnight in PBS containing 0.1% Tween20 and 4% dry powdered milk. After washing, a secondary alkaline phosphatase-conjugated goat anti-mouse antibody at 1:5000 was added for 2 hr. The bound antibodies were detected by reacting with substrate solution containing 5-bromo-4-cholor-indolyl-phosphase and 4-Nitro Blue Tetrazolium chloride.

## Immunohistochemistry

Germarium preparations for whole mount immunofluorescence were prepared as according to (*Page and Hawley, 2001*) with minor exceptions. Three- to five-day-old females were collected and yeasted overnight in the presence of males. Ovaries were dissected in PBS for no longer than 20 min prior to fixing (200 μL of PBS containing 2% formaldehyde (Ted Pella, Redding, CA) and 0.5% Nonidet P-40 plus 600 μL heptane) at room temperature. Ovaries were then washed three times for 10 min in PBS with 0.1% Tween (PBST). Late stage egg chambers were removed by cutting the ovaries with forceps, and the ovarioles containing the germarium tips were teased apart before being blocked in PBST with 1% bovine serum albumin (BSA) (EMD Chemicals, San Diego, CA) for one hr. Primary antibody diluted in PBST was incubated with germarium tips overnight at 4°C while nutating. After washing three times for 20 min in PBST the secondary antibodies were added for 4 hr followed by the addition of 4'6-diamididino-2-phenylindole (DAPI) at a concentration of 1μg/ml for the final 10 mins of incubation. Ovary material was washed as before and the samples were mounted in Pro-Long Gold (Life Technologies, Grand Island, NY).

For immuno-EM samples, three- to five-day-old mated females were yeasted overnight. Ovaries from five to seven females were dissected in cold Ringer's solution and fixed at 20°C for 30 min (inverting every 10 min) in 200 μL of PBS containing 3% EM-grade formaldehyde (Ted Pella, Redding, CA) and 0.5% Nonidet P-40, plus 600 μL of hexane. Ovaries were washed in PBST as above and quenched with fresh 0.1 M ammonium chloride in PBS. Following another set of washes, ovaries were blocked for 1 hr at RT in 1% BSA in PBST. Primary antibodies were added and incubated in 1% BSA PBST overnight at 4° C. The ovaries were washed 6 x 10 min each in PBST and incubated with secondary for 4 hr in 1% BSA, 0.1% cold water fish gelatin (Electron Microscopy Sciences, Hatfield, PA), and 2% normal goat serum in PBST. Secondary antibodies used were anti-rabbit Alexa-488 and anti-rat ultra-small gold (Electron Microscopy Sciences, Hatfield, PA). After washing 6 x 10 min, ovaries were post-fixed as before except at RT. Ovaries were then washed in distilled water for 3 x 20 min and gold was enhanced with Aurion R-GENT SE-EM (Electron Microscopy Sciences, Hatfield, PA) for 1 hr 15 min. Following silver enhancement, samples were treated with 0.03 M sodium thiosulfate in distilled water for 5 min, followed by 3 x 10 min washes with distilled water. Ovaries were then post-fixed in 1% $OsO_4$ in PBS for 30 min at RT, washed as before with water and dehydrated in ethanol. Samples were embedded in epoxy resin at RT for two days followed by polymerization at 60° for two days. Serial sections (50 nm thick) were cut and transferred to formvar-carbon-coated slot grids and stained with aqueous uranyl acetate and lead citrate. Sections that contained SC were first identified by immunofluorescence, and those sections were then imaged on an FEI transmission electron microscope (80 kv).

Primary antibodies used include mouse anti-C(3)G 1A8-1G2 (1:500) (*Anderson et al., 2005*), affinity-purified rabbit anti-Corolla (animal 210) (1:2000) (*Collins et al., 2014*), rat anti-CID (1:2000) (gift of Sunkel Laboratory) (*Martins et al., 2009*), mouse anti-Orb antibodies 4H8 and 6H4 (1:40 each) (Developmental Studies Hybridoma Bank, Iowa), mouse anti-γ-H2AV (1:1000) (Lake et al., 2013), mouse anti-HA.11 (Covance, Princeton, NJ), and high-affinity rat anti-HA clone 3F10 (1:100 IF or 1:50 immuno-EM) (Roche, Indianapolis, IN). Secondary goat anti-mouse, rabbit or rat Alexa-488, Alexa-555 and Alexa-647 IgG H&L chain conjugated antibodies were used at 1:500 (Molecular Probes, Life Technologies, Grand Island, NY), and secondary goat anti-rat ultra-small gold IgG H&L chain conjugated antibody (1:50) (Electron Microscopy Sciences, Hatfield, PA).

## Microscopy and image analysis

Images were acquired with a DeltaVision microscopy system (GE Healthcare, Piscataway, NY) consisting of a 1x70 inverted microscope with a high-resolution CCD camera or an Applied Precision OMX Blaze microscope (Issaquah, WA, USA) equipped with a PCO Edge sCMOS camera. Images were deconvolved (DeltaVision and OMX) and reconstruction was performed (OMX) using SoftWoRx v. 6.1 software (Applied Precision/GE Healthcare) following Applied Precision protocols.

To analyze the specificity of the rat HA antibody for Vilya$^{3XHA}$ protein we compared staining of anti-HA on *vilya$^{3XHA}$*-expressing tissue to staining on wild-type tissue. We prepared samples for each in parallel. We acquired five germarium and five late stage images of *vilya$^{3XHA}$*-expressing tissue using a target intensity value of 3000 on the DeltaVision microscopy system for each filter (DAPI, TRITC (anti-HA) and Cy5 (anti-Corolla). We recorded the percent transmission (which stayed consistent) and exposure time in each channel for each of the acquired images. We averaged the five exposure times for each filter. We fixed these as values for our wild-type image acquisition. The images were then deconvolved as above. In addition, we recorded the exposure time on wild type for each of the filters when using a target intensity value of 3000. These were then averaged as before and compared to the averages of *vilya$^{3XHA}$*-expressing tissue as a ratio of average exposure time in *vilya$^{3XHA}$*-expressing ovaries to average exposure time in wild type. For the DAPI the ratio was 1.0:0.9, TRITC (HA) 1.0:5.78 and Cy5 (Corolla) 1.0:1.2. Thus in order to reach the same intensity value, the exposure time had to be increased on wild-type tissue by almost six times.

The analysis of centromere clustering was performed as previously described (*Takeo et al., 2011*), where individual oocyte nuclei were scored for the number of CID foci by analyzing CID staining throughout each section of the nucleus in SoftWoRx.

To determine the number of γH2AV foci or Vilya$^{3XHA}$ foci, we used Imaris software 7.7.2 (Bitplane, Zurich, Switzerland) to crop in 3D each oocyte using the SC to define the sections pertaining to each nucleus. Using Imaris software, we displayed each z-section using the gallery function and only clearly defined foci were counted manually in the corresponding z-series.

To determine the colocalization frequency of γH2AV and Vilya$^{3XHA}$ foci, we performed 3D crop of the selected nuclei as above. We identified the number of Vilya$^{3XHA}$ foci for each nucleus analyzed as described above. We then rotated the images in 3D to verify that the γH2AV signal was not simply above or below the Vilya$^{3XHA}$ focus. We scored those signals that overlapped, as well as those signals that were adjacent (no apparent gap between the foci), but not separated above or below in 3D, as being associated. To verify the relevance of their association, we took each 3D cropped oocyte nucleus and rotated the channel for the Vilya$^{3XHA}$ foci by 180 degrees using ImageJ software. First, we split each of the channels of the image. We selected the channel with the Vilya$^{3XHA}$ foci, and used the transform function to flip the z-series horizontally. We then used the stack tool to reverse the stack. Together these two manipulations are equivalent to a 180 degree rotation of the Vilya$^{3XHA}$ channel. After merging the channels back together, we again analyzed the association of γH2AV and Vilya$^{3XHA}$ foci as before.

3D projections and tracing of SC between homologous chromosome arms was performed using Imaris software, and maximum intensity projections were made unless otherwise noted. Image J custom plugins for straightening of Imaris spot profiles are available at http://research.stowers.org/imagejplugins.

## X-ray treatment

For immunofluorescence analysis of DSBs created by X-ray, three- to five-day-old mated females were exposed to 1000 rad of X-ray at a dose of 112 rad/min. Ovaries from treated (or non-treated control females) were collected and fixed as above 5 hr after X-ray treatment.

## Fluorescence in situ hybridization

Ovaries from three- to five-day mated females that had been yeasted for one day were dissected. FISH and immunofluorescence was performed as previously described (*Blumenstiel et al., 2008*) using amine-labeled probes made with ARES Alex Fluor DNA labeling kit (Invitrogen Life Technologies, Grand Island, NY) for euchromatic region *14*. Overlapping region 14 BACs were labeled and used (*BACR03G18, BACR06P10* and *BACR13G13*) (CHORI). Pairing was determined as previously described (*Page et al., 2008*; *Joyce et al., 2013*).

## Acknowlegments

We would like to thank the Hawley Laboratory for helpful advice on the manuscript, especially Amanda Bonner for assistance with sequence alignment. We would also like to thank Zulin Yu for assistance with the OMX microscope, and Melainia McClain for EM assistance. RSH is supported by Stowers Institute for Medical Research and is an American Cancer Research Professor supported by the award RP-05-086-06DDC.

## Additional information

### Funding

| Funder | Grant reference number | Author |
|---|---|---|
| Stowers Institute for Medical Research | | R Scott Hawley |
| American Cancer Research | RP-05-086-06DDC | R Scott Hawley |

The funders had no role in study design, data collection and interpretation, or the decision to submit the work for publication.

### Author contributions

CML, Conception and design, Drafting or revising the article; RJN, Acquisition of data, Analysis and interpretation of data, Drafting or revising the article; FG, JRU, BDS, Acquisition of data, Analysis and interpretation of data; RSH, Conception and design, Acquisition of data, Analysis and interpretation of data, Drafting or revising the article

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
