## [Decision Letter]

Thank you for submitting your work entitled "Vilya, a component of the
recombination nodule, is required for meiotic double-strand break formation in
Drosophila" for peer review at *eLife*. Your submission has been
favorably evaluated by Detlef Weigel (Senior editor) and three reviewers, one of whom is
a member of our Board of Reviewing Editors.

The reviewers have discussed the reviews with one another and the Reviewing editor has
drafted this decision to help you prepare a revised submission.

Based on a previous germline mutagenesis screen (Collins et al., 2012) and a
complementation assay (this study), the authors identified a gene named Vilya, involved
in meiotic double-strand break (DSB) formation in *Drosophila*. The use
of Î³H2AV antibody developed in an earlier study (Lake et al., 2013) has allowed the
authors to quantify DSB formation and conclude that a truncation of the Vilya gene
causes defect in DSB formation. The protein shares some features characteristic of the
yeast Zip3 protein family, important for crossover designation. Vilya interacts with the
DSB accessory protein Mei-P22, is localized in oocytes along the synaptonemal complex,
and forms foci that appear to be recombination nodules by electron microscopy.
Altogether, the results suggest that Vilya plays an important role in DSB formation
and/or repair, and help to provide a more complete model of meiotic recombination
pathway in *Drosophila* females, where several recombination factors well
conserved among eukaryotes appear to be absent.

Overall, the data quality are excellent and the writing is clear. The discovery of Vilya
is important for several reasons: it is the first (likely) member of the Zip3/Rnf212
family discovered in *Drosophila*, it is the first protein component
clearly demonstrated for *Drosophila* recombination nodules, and it is
the first example in any species of a role for a protein in this family in promoting DSB
formation.

However both the proposed roles of Vilya in DSB formation and in crossover control need
to be further substantiated: Additional experiments to further evaluate the implication
of Vilya for DSB formation and to determine its relationship with C(3)G and to
demonstrate a role downstream of DSB formation should solve these issues. Some
straightforward control experiments and clarifications in text and figures would
strengthen the paper.

Important revisions:

Questions related to the role of Vilya for DSB formation:

1) It is not entirely clear if one can conclude that no DSB are formed in the absence of
Vilya. In the subsection “*vilya* is required for programmed DSB
formation in early pachytene”, the authors state: "Immunofluorescence analysis of
early pachytene oocytes reveals a complete failure to initiate programmed DSBs". It
would be better to use more nuanced language: This experiment suggests a failure or a
strong reduction, and the resolution of the picture does not allow for the conclusion
"complete" failure. What is the estimated limit of sensitivity of the gH2AV
staining? The data of the recombination map is clear but if Vilya is required for CO,
then some remaining DSB could be repaired to NCO in the mutant.

2) Is Vilya required for DSB formation in nurse cells? This is quite important and would
allow to distinguish between a direct role for DSB formation and a regulatory role as
observed for C(3)G.

3) Also, an alternative interpretation, may be not likely, but that should be mentioned
is that the mutant phenotypes could be explained by assuming a fast repair of DSB to NCO
(or sister) in Vilya mutant and not requiring Rad54.

4) The interplay between Vilya and C(3)G should be better documented: Is Vilya
localization dependent on C(3)G? If yes, how could one explain the DSB activity detected
in C(3)G mutant? What is the DSB level in Vilya C(3)G double mutant?

5) An important test for the deficiency of DSB formation is the suppression of okra
mutant phenotypes. One missing piece of information is the detection of gH2AV in vilya
okra double mutant (region 2A and region 3).

6) The interpretation of the ring mutations on the interaction with MeiP-22 is not
convincing as the mutant protein may have an improper folding or conformation, this
alternative should be taken into account in the Discussion. Western blots should be
provided as well as two hybrid data for the RING domain mutants.

7) The colocalization of Vilya with gH2Av is not convincing. If Vilya foci mark CO, one
does not expect a high colocalization frequency between gH2AV and Vilya (may be only 30%
or less). In addition the protocol used for image analysis and evaluating colocalization
is poorly described. What does closely associated mean? What does adjacent mean? How is
this different from expected by chance? Randomization controls should be performed to
evaluate the degree of overlap, e.g., by flipping one of the fluorescence channels by
180° and re-evaluating the degree of fortuitous overlap. In the third paragraph of the
subsection “The formation of discrete Vilya^3XHA^ foci is dependent on
programmed DSB formation”: The difference in Vilya-gH2AV association does not appear to
be significant (Fisher's exact test gives p = 0.175). Please clarify.

8) In terms of Vilya foci quantification, it is not clear what the parameters to
identify a focus are, in particular taking into account that their intensity varies and
that there is signal in the central region. For instance in Figure 6, there seems to be more than 12 Vilya foci in the
nucleus.

*Questions related to the role of Vilya for Crossover control:*

9) To firm up the speculation about a possible late role for Vilya and to reconcile the
cytology and the phenotypic analysis, do X-rays rescue Vilya foci in a mei-P22 or
mei-W68 mutant? This experiment would help clarify whether Vilya has functions
downstream of DSB formation, as speculated.

10) In the first paragraph of the subsection “Vilya's role in DSB formation and crossing
over” and Figure 8: The authors' model is not
convincing as spelled out. If Mei-W68 catalyzes DSBs at Mei-P22 sites, and if Vilya is
then recruited to a subset of DSB sites, one should expect to see more Mei-P22 foci than
Vilya foci. This data is missing in the manuscript and is important to support the model
in Figure 8. Furthermore, in Liu et al., 2002,
the average number of Mei-P22 foci per SC containing cell in region 2A was calculated at
8.7, which is not significantly higher than the average number of Vilya3XHA foci in
early pachytene region 2A calculated at 8.0 in the present study (Figure 3). Moreover, the poor colocalization (13,5%) observed
between Î³H2AV and MEI-P22 foci in Mehrotra & Mckim, 2006 doesn't fit with MEI-P22
being a mark of future DSB sites. The manuscript would be significantly strengthened if
these issues could be addressed. Why not integrate trem in the model?

11) Figure 3: As Vilya3XHA shows two distinct
types of staining (linear and foci), the specificity of the anti-HA antibody should be
shown in a control experiment (e.g. anti-HA on a WT oocyte).

12) Information about the level of expression of the transgene (relative to endogene)
expressing tagged Vilya is needed. Possible artefacts due to overexpression should be
discussed for interpreting the localization of Vilya.

---

## [Author Response]

Questions related to the role of Vilya for DSB formation:

*1) It is not entirely clear if one can conclude that no DSB are formed in the
absence of Vilya. In the subsection “*vilya *is required for
programmed DSB formation in early pachytene”, the authors state:
"Immunofluorescence analysis of early pachytene oocytes reveals a complete
failure to initiate programmed DSBs". It would be better to use more nuanced
language: This experiment suggests a failure or a strong reduction, and the
resolution of the picture does not allow for the conclusion "complete"
failure. What is the estimated limit of sensitivity of the gH2AV staining? The data
of the recombination map is clear but if Vilya is required for CO, then some
remaining DSB could be repaired to NCO in the mutant.*

We now use more nuanced language throughout the manuscript, suggesting that there is a
strong reduction rather than asserting that there is a complete elimination of DSB
formation during pachytene. In support of our assertion that there is a very strong
reduction in crossing over, we have also added a recombination analysis for the entire
*3rd* chromosome to complement the *X* chromosome
recombination data already included (see Figure 2—figure supplement 2). These new data show that the frequency of
recombination on the *3rd* chromosome in
*vilya^826^* is reduced over 50-fold compared to wild type.
In addition, we have now included DSB numbers for the transheterozygote
(*Df/vilya^826^*). Together with
the data already included in the manuscript, these experiments show that there is a very
strong reduction in both the initiation of DSBs and in crossing over.

*2) Is Vilya required for DSB formation in nurse cells? This is quite important
and would allow to distinguish between a direct role for DSB formation and a
regulatory role as observed for C(3)G.*

We have included data regarding the lack of induction of DSBs in nurse cells in the
*vilya* mutant in a new figure (Figure 1—figure supplement 4). In this figure we show the pattern of DSB
induction within all nuclei in region 2A for both wild type and *c (3)G*
(which shows a pattern similar to wild type). However, we show that *mei-W68,
mei-P22, vilya^826^* and *Df/vilya^826^* all
exhibit a strong reduction in the initiation of DSBs in both nurse cells and oocyte
nuclei. These images further demonstrate that Vilya has a general role in DSB formation,
similar to Mei-W68 and Mei-P22, rather than an oocyte-specific regulatory role in DSB
formation, as has been reported for C(3)G (Mehrotra and McKim, 2006).

*3) Also, an alternative interpretation, may be not likely, but that should be
mentioned is that the mutant phenotypes could be explained by assuming a fast repair
of DSB to NCO (or sister) in Vilya mutant and not requiring Rad54.*

We have added this as a possibility in the subsection “Meiosis in
*Drosophila*”.

*4) The interplay between Vilya and C(3)G should be better documented: Is Vilya
localization dependent on C(3)G? If yes, how could one explain the DSB activity
detected in C(3)G mutant?*

We have now included data to show that the localization of Vilya to discrete foci is not
dependent on C(3)G. However no linear element Vilya^3XHA^ staining was observed
in the *c (3)G* mutant. Our data show that 75% of the
Vilya^3XHA^ foci observed in region 2A oocytes colocalize/associate with the
Î³H2AV marks (see new Figure 7). However, very
few region 2B oocyte nuclei retained any Vilya^3XHA^ foci. We speculate that
the inability to retain Vilya^3XHA^ at discrete foci is the result of the
failure to convert the DSBs that do form into crossovers – a process known to require
the SC. We have added these observations to the subsection “The formation of discrete
Vilya^3XHA^ foci is dependent on programmed DSB formation but not the SC” in
the Results section.

*What is the DSB level in Vilya C(3)G double mutant?*

Previous correspondence with the editor led to an agreement that the analysis of the
*vilya; c (3)G* double mutant, while interesting, was beyond the scope
of this study.

*5) An important test for the deficiency of DSB formation is the suppression of
okra mutant phenotypes. One missing piece of information is the detection of gH2AV in
vilya okra double mutant (region 2A and region 3).*

Again, previous discussions with the editor regarding measuring the DSB level in the
*vilya; okra* double mutant led to the agreement that such experiments
could be deferred to a future paper.

*6) The interpretation of the ring mutations on the interaction with MeiP-22 is
not convincing as the mutant protein may have an improper folding or conformation,
this alternative should be taken into account in the Discussion. Western blots should
be provided as well as two hybrid data for the RING domain mutants.*

We agree that the failed interaction between Mei-P22 and the Vilya RING domain point
mutants may be the result of improper protein folding or confirmation. We have now added
this possibility to the Results section. We have also provided the yeast two-hybrid data
and the Western blot as a new figure (Figure 9—figure supplement 1). In addition, we have added a new Materials and methods section
regarding this experiment in the yeast two-hybrid section, as well as a figure legend
for the new figure.

*7) The colocalization of Vilya with gH2Av is not convincing. If Vilya foci mark
CO, one does not expect a high colocalization frequency between gH2AV and Vilya (may
be only 30% or less). In addition the protocol used for image analysis and evaluating
colocalization is poorly described. What does closely associated mean? What does
adjacent mean? How is this different from expected by chance? Randomization controls
should be performed to evaluate the degree of overlap, e.g., by flipping one of the
fluorescence channels by 180° and re-evaluating the degree of fortuitous overlap. In
the third paragraph of the subsection “The formation of discrete Vilya^3XHA^
foci is dependent on programmed DSB formation”: The difference in Vilya-gH2AV
association does not appear to be significant (Fisher's exact test gives p = 0.175).
Please clarify.*

We have added more details regarding the process of image analysis to the Materials and
methods. Our definition of adjacent simply means no apparent gap between the foci. In
addition, as suggested, we performed randomization controls (see subsection “The
formation of discrete Vilya^3XHA^ foci is dependent on programmed DSB formation
but not the SC” of Results and subsection “Microscopy and image analysis “of Materials
and methods) and have now included them in the Results and Methods sections. Since we
wanted to be able to perform the randomization controls on all oocytes scored for
colocalization, we used only images that contained one oocyte – many region 2A oocytes
cannot be individually and completely cropped due to their proximity to another
SC-containing nucleus, even though you can clearly distinguish between the two nuclei in
3D. This is why the numbers in the Results section regarding this data have changed in
the revised version. The overall association of Î³H2AV and Vilya^3XHA^ foci
changed only slightly from 58% to 60.5% in scoring these 11 nuclei.

We have also attempted to better explain the uncertainties that surround the
quantitative relationship between the numbers of Vilya and Î³H2AV foci in terms of
laying the groundwork for the model presented in the Discussion of the paper. As Vilya
is required for DSB formation (a dynamic process) and is also likely marking CO sites
which no longer bear the Î³H2AV mark, it is more difficult to predict the frequency of
association that would be predicted in region 2A when DSBs are being induced. We do know
that the number of Vilya^3XHA^ foci in region 2B is consistent with that of
RNs/COs.

*8) In terms of Vilya foci quantification, it is not clear what the parameters to
identify a focus are, in particular taking into account that their intensity varies
and that there is signal in the central region. For instance in Figure 6, there seems to be more than 12 Vilya foci in the
nucleus.*

We understand how the reviewer came to the conclusion that there were more than 12 foci
in Figure 6. In this particular image, a few
Vilya foci are not in the main nucleus shown. This is because it is sometimes difficult
to crop out a region 2A nucleus in such a manner as to be free of sections from
neighboring nuclei. In region 2A, up to four neighboring nuclei contain full-length SC
and enter meiosis. Although you can clearly see the boundary in 3D, upon projection of
the entire stack, these nuclei looked connected, when in fact they are not. In
retrospect we should have chosen a better image that was able to be cropped and
displayed as information from a single nucleus. We have now changed that image and also
added in the Materials and methods section more information on foci quantification (for
both Vilya and DSBs).

Questions related to the role of Vilya for Crossover control:

*9) To firm up the speculation about a possible late role for Vilya and to
reconcile the cytology and the phenotypic analysis, do X-rays rescue Vilya foci in a
mei-P22 or mei-W68 mutant? This experiment would help clarify whether Vilya has
functions downstream of DSB formation, as speculated.*

We were able to include the experiment suggested by the reviews in analyzing whether
DSBs created by X-ray can recruit Vilya to them as discrete foci. The short answer is
that “yes” X-ray-induced breaks can induce Vilya to form foci. We have included that new
data as a new figure (Figure 8).

*10) In the first paragraph of the subsection “Vilya's role in DSB formation and
crossing over” and Figure 8: The authors'
model is not convincing as spelled out. If Mei-W68 catalyzes DSBs at Mei-P22 sites,
and if Vilya is then recruited to a subset of DSB sites, one should expect to see
more Mei-P22 foci than Vilya foci. This data is missing in the manuscript and is
important to support the model in Figure 8.
Furthermore, in Liu et al., 2002, the average number of Mei-P22 foci per SC
containing cell in region 2A was calculated at 8.7, which is not significantly higher
than the average number of Vilya3XHA foci in early pachytene region 2A calculated at
8.0 in the present study (Figure 3).
Moreover, the poor colocalization (13,5%) observed between Î³H2AV and MEI-P22 foci in
Mehrotra & Mckim, 2006 doesn't fit with MEI-P22 being a mark of future DSB sites.
The manuscript would be significantly strengthened if these issues could be
addressed. Why not integrate trem in the model?*

As we pointed out in our response to reviewer concern number 6, we were not able to
obtain additional information regarding the localization of Mei-P22 during the revision
process. Therefore, we have chosen to produce a minimalistic model with the information
we currently have. As we only ever see a fraction of the DSBs made in region 2A using
the antibody to Î³H2AV (the only marker we have in flies), we cannot predict what the
expected number of Mei-P22^3XHA^ or Vilya^3XHA^ foci should be in this
region (see above) compared to the number of Î³H2AV foci. We can only report on the
number of foci per stage of meiosis and their overall trend throughout the germarium. We
have also removed the suggestion that Mei-P22 is marking future DSB sites.

Although we would like to eventually incorporate Trem into the model, the null allele of
*trem* doesn't allow us to look at localization of Vilya in this
mutant. This is because the *P*-element insertion in
*trem* is a *pBac[WH]* element that contains a UAS in
the 5' UTR of the *trem* gene. Therefore, expression of Vilya using the
GAL4-UAS system in this *trem* null allele also expresses
*trem*. In the future we will need to make either a tagged germline
*vilya* expression construct for this analysis or use CRISPR to tag
the enodogenous *vilya* gene, as we have been completely unsuccessful at
making a Vilya antibody after numerous attempts (three additional attempts even during
this revision process). Therefore, until we are able to fully analyze their relationship
or get preliminary data, we have chosen to leave Trem out of the model for now.

*11) Figure 3 Vilya3XHA shows two
distinct types of staining (linear and foci), the specificity of the anti-HA antibody
should be shown in a control experiment (e.g. anti-HA on a WT oocyte).*

We have added a control image of the rat anti-HA antibody as part of Figure 3—figure supplement 1. In Figure 3—figure supplement 1 we show a region 2A
nucleus that has both types of staining (where the foci predominate) and a later stage
(stage 4) showing the specificity to the linear staining. (Stage 4 is a stage between
the region 3 and the stage 6 already shown in Figure 3—figure supplement 1.) In both cases the controls show insignificant
staining levels. The remaining panels in this figure have been moved from B and C to C
and D. We have also added details regarding how the control images were collected in the
Materials and methods and have pointed to this location in the figure legend.

*12) Information about the level of expression of the transgene (relative to
endogene) expressing tagged Vilya is needed. Possible artefacts due to overexpression
should be discussed for interpreting the localization of Vilya.*

We mention in the manuscript that we are using an overexpression system that is highly
expressed throughout the ovariole. We also have been very aware to use throughout the
manuscript the 3XHA reference when discussing the localization of Vilya, as to remind
the reader this is a tagged overexpressed construct. We have included in the manuscript
caveats to using an overexpression construct for certain experiments as well.

We have not included levels of expression of the transgene for the following reasons: 1)
We know and mention that in using the *nos*-GAL4-UAS system, we are
highly expressing Vilya at all stages of oogenesis. 2) The nurse cells undergo many
rounds of endoreduplication leading to hundreds of copies of the genome per each of the
15 nurse cells. These nurse cells synthesize both protein and RNAs, which are dumped
into the oocyte. We would be unable to determine whether a certain level of increase in
mRNA resulted in that level of protein expression, as well due to the inability to
isolate germarium, we are unable to look at expression levels specifically in the area
of the ovary we are studying in this manuscript.